# Demographically-Informed Prediction Discrepancy Index: Early Warnings of Demographic Biases for Unlabeled Populations

Lucas Mansilla     *lmansilla@sinc.unl.edu.ar*
Estanislao Claucich     *eclaucich@sinc.unl.edu.ar*
Rodrigo Echeveste     *recheveste@sinc.unl.edu.ar*
Diego H. Milone     *dmilone@sinc.unl.edu.ar*
Enzo Ferrante     *eferrante@sinc.unl.edu.ar*

*Research Institute for Signals, Systems and Computational Intelligence - sinc(i)*
*Universidad Nacional del Litoral - CONICET, Santa Fe, Argentina*

**Reviewed on OpenReview:** *https://openreview.net/forum?id=8W6IDyFZgC*

## Abstract

An ever-growing body of work has shown that machine learning systems can be systematically biased against certain sub-populations defined by attributes like race or gender. Data imbalance and under-representation of certain populations in the training datasets have been identified as potential causes behind this phenomenon. However, understanding whether data imbalance with respect to a specific demographic group may result in biases for a given task and model class is not simple. An approach to answering this question is to perform controlled experiments, where several models are trained with different imbalance ratios and then their performance is evaluated on the target population. However, in the absence of ground-truth annotations at deployment for an unseen population, most fairness metrics cannot be computed. In this work, we explore an alternative method to study potential bias issues based on the output discrepancy of pools of models trained on different demographic groups. Models within a pool are otherwise identical in terms of architecture, hyper-parameters, and training scheme. Our hypothesis is that the output consistency between models may serve as a proxy to anticipate biases concerning demographic groups. In other words, if models tailored to different demographic groups produce inconsistent predictions, then biases are more prone to appear in the task under analysis. We formulate the Demographically-Informed Prediction Discrepancy Index (DIPDI) and validate our hypothesis in numerical experiments using both synthetic and real-world datasets. Our work sheds light on the relationship between model output discrepancy and demographic biases and provides a means to anticipate potential bias issues in the absence of ground-truth annotations. Indeed, we show how DIPDI could provide early warnings about potential demographic biases when deploying machine learning models on new and unlabeled populations that exhibit demographic shifts.

## 1 Introduction

Machine learning (ML) models are susceptible to exhibiting biases against certain subpopulations defined in terms of sensitive demographic characteristics such as gender, age, or race. Examples of such biases can be found in a variety of fields, including predictive policing (Angwin et al., 2016), facial analysis (Buolamwini & Gebru, 2018), and healthcare (Chen et al., 2019; Ricci Lara et al., 2022). Factors that contribute to biased models may include the data used for training and evaluation, as well as decisions made during the development process (Suresh & Guttag, 2019). As ML applications in the real world become increasingly

widespread, it is important to evaluate models to ensure that they are not only accurate but also produce fair and ethical results.

In particular, under-representation of certain demographic groups has been identified as one of the main causes of bias when developing predictive systems. Although many types of biases exist and can be measured using different metrics, here we are mostly concerned about disparities in predictive performance usually studied in the literature of group fairness (e.g. as measured by the gap in accuracy between different demographic groups for classification systems, or the gap in mean absolute error per demographic group for regression problems). For example, gender imbalance in X-ray medical imaging datasets has been shown to have a significant impact on the performance of assisted diagnosis systems for thoracic diseases based on convolutional neural networks (Larrazabal et al., 2020), as measured by the gap in area under the receiver operating curve (AUC-ROC) for male and female individuals. Another example is given by under-representation of ethnic groups, which has also been found to influence model performance for cardiac image segmentation (Puyol-Antón et al., 2021), as measured by the differences in the Dice coefficient between different groups. However, in other tasks, such data imbalance has not been associated with unequal performance. In Petersen et al. (2022) for example, the authors found that in the case of Alzheimer's disease prediction from brain magnetic resonance images (MRI), gender imbalance in the training dataset did not lead to a clear pattern of improved model performance for the majority group. A similar phenomenon was observed in Kinyanjui et al. (2020), where the authors studied under-representation of skin color when analyzing dermoscopic images for skin cancer detection, and did not observe such disparities. This observation was then challenged by Groh et al. (2021), which found disparities in performance arising from training a neural network on only a subset of skin types. In all, it is not always a fact that data imbalance will result in biased automated systems. To complicate matters further, even when the presence of biases can be assessed during the development of an automated tool, these properties may not transfer under distribution shifts (Schrouff et al., 2022), for instance, once the model is deployed. This is a problem for fairness metrics which require ground-truth annotations, which are expensive to obtain and may not be available before deployment. Given these issues, a valid question that one may then ask is: can we anticipate whether models will exhibit biases with respect to data imbalance in terms of a particular protected attribute in the absence of ground-truth annotations?

Typical approaches to identify biases in ML models involve subgroup analysis and controlled experiments where both demographic and target labels are available (Larrazabal et al., 2020; Buolamwini & Gebru, 2018; Glocker et al., 2021). Model performance across demographic groups is commonly evaluated employing one or more metrics (Corbett-Davies & Goel, 2018) with the implicit assumption that the presence or absence of biases during development will be representative of the behaviour of these models when applied to previously unseen data at deployment. Recent findings regarding how fairness properties transfer across distribution shifts in real-world healthcare applications due to changes in geographic location or population demographics, warn us about the risks of this assumption (Schrouff et al., 2022). A system that did not exhibit strong biases in the source population may begin to do so when the target population changes. This is particularly concerning in applications like healthcare, where collecting expert annotations on large datasets can be costly and time-consuming (Ricci Lara et al., 2022), meaning that fairness metrics requiring labels may not be computed, with the result of biases going unnoticed. In this context, developing methods that can be used without the need for ground truth in the target population becomes highly relevant. In this paper, we are interested in exploring ways to anticipate potential bias issues that may arise in the context of a given task for a novel unlabeled target population. We do so by proxy: using an index that we call Demographically-Informed Prediction Discrepancy Index (DIPDI), which can be computed in the absence of ground truth annotations. We provide an analytical derivation demonstrating the relation between DIPDI and performance gaps, and show in numerical experiments using both synthetic and real-world datasets that this index is indeed indicative of bias proneness, providing an early warning for potential fairness issues in these settings.

## 2 Related work

The implicit assumption that model assessment during development is representative of its behaviour at deployment is not unique to fairness studies. Indeed, anticipating whether a model will systematically fail

or not when ground-truth annotations are not available is a current topic of interest in the field, and one way to tackle this issue is to look at predictive uncertainty (Gal et al., 2016). Intuitively, if a well-calibrated model systematically makes highly uncertain predictions for certain individuals, then chances are that these predictions will have a higher failure rate for those individuals. In this context, recent studies have analyzed the relation between fairness and uncertainty, postulating that uncertainty estimates can be used to obtain fairer models, improve decision-making, and build trust in automated systems (Bhatt et al., 2021). For example, Lu et al. (2021) analyzed how alternative uncertainty estimation methods can be used to evaluate subgroup disparities in mammography image analysis, while Stone et al. (2022) leveraged epistemic uncertainty estimates to mitigate minority group biases during training. The work of Dusenberry et al. (2020) discusses the role of model uncertainty in predictive models for Electronic Health Record (EHR), and shows how it can change across different patient subgroups, in terms of ethnicity, gender and age, considering Bayesian and deep ensemble approaches for uncertainty estimation. Even though in this work we do not directly rely on the notion of uncertainty, our study is highly influenced by this idea, as it explores the use of output discrepancy for a set of models as a way of anticipating bias issues. This notion is closely related to ensemble variance, usually employed as a measure of uncertainty for ensemble methods (Lakshminarayanan et al., 2017; Pividori et al., 2016; Larrazabal et al., 2021). Another important concept in our study is that of consistency (Wang et al., 2020), defined as the ability of a set of multiple trained learners to reproduce an output for the same input. According to this concept, model outputs are analyzed irrespective of whether they are correct or incorrect, and as such, it does not require ground-truth annotations to be computed. This idea will be central to our study, as we explore how changes in consistency for pools of models trained on the same or different demographic groups will correlate with potential biases that may emerge in a given task.

**Contributions:** Here we present a methodology to understand whether biases with respect to a given demographic attribute are prone to arise in a new unlabeled dataset. We do so by analyzing the output consistency of a pool of models, where each model is *trained on separate demographic groups*, but is otherwise identical in terms of architecture, hyper-parameters and training scheme. We introduce a new index, DIPDI, based on the following hypothesis: if models specialized in different demographic groups produce discrepant predictions for the same test data, then the task under analysis is prone to be biased against that demographic attribute. Note that throughout this manuscript, we consider that a task is prone to be biased with respect to a given demographic attribute when we observe systematic performance gaps for models trained on different demographic groups characterized by such attribute.

We validate our hypothesis using synthetic and real-world datasets, focusing on regression and classification tasks: age regression from face photos and X-ray images, classification of younger vs older celebrities in face images, as well as hair color classification. We use four real-world datasets and consider different cases of demographic imbalance in the training data. Our results indicate that DIPDI can be used to anticipate potential bias issues in the absence of ground truth labels, and confirm the association between output discrepancy and bias proneness. We also assess the behaviour of DIPDI for unseen populations with different types of distribution shifts, showing how it can be used to measure bias proneness in dynamic contexts. Moreover, since our metric does not require expert annotations to be computed, it could help to anticipate bias issues in real-world scenarios and give early warnings when deploying machine learning models on new, unlabeled populations.

# 3 Demographically-Informed Prediction Discrepancy Index (DIPDI)

## 3.1 Quantifying output discrepancy within and between demographically-informed sets of models

Given two sets of predictive models $\mathbb{A} = \{A_1, A_2\}$ and $\mathbb{B} = \{B_1, B_2\}$, we are interested in analyzing how the output discrepancy of models within the same set compares to the output discrepancy of models coming from different sets, when they are evaluated on samples from an unlabeled dataset $\mathbb{D}$. Here $A(\mathbf{x}_k) : \mathbb{X} \longrightarrow \mathbb{Y}$ is a predictive model (e.g. a regression or classification model), where $\mathbf{x}_k \in \mathbb{D} \subseteq \mathbb{X}$ can be images or other types of data for subject $k$, and the output of $A(\mathbf{x}_k)$ is a label $y \in \mathbb{Y}$ which could be a real number for regression problems as well as a categorical label or a soft probability estimate for classification problems.

We then define an *average output discrepancy* function $\mathcal{N}_{\mathbb{D}}(M_1, M_2)$, that takes as input two models $M_1$ and $M_2$, and returns a number representing how different their outputs are on average when evaluated on all samples from $\mathbb{D}$. We measure the discrepancy between two models using a discrepancy function $d(\cdot, \cdot)$ so that the average output discrepancy is defined as

$$\mathcal{N}_{\mathbb{D}}(M_1, M_2) = \frac{1}{|\mathbb{D}|} \sum_{\mathbf{x}_k \in \mathbb{D}} d(M_1(\mathbf{x}_k), M_2(\mathbf{x}_k)). \tag{1}$$

For example, in regression problems the discrepancy function $d(\cdot, \cdot)$ could be the absolute or the quadratic error, while in classification problems it could be the Jensen–Shannon divergence between the output distributions for models $M_1, M_2$. In other words, the average output discrepancy is the mean discrepancy $d(M_1, M_2)$ between the predicted values of models $M_1$ and $M_2$ for all subjects in the dataset. It returns a number closer to 0 when the outputs of the two models *for every data sample* are similar, and higher if they tend to differ. Since we are interested in analyzing the *output discrepancy* for models within and between sets, we consider the following ratio as an indicator of relative output discrepancy:

$$\Phi_{\mathbb{D}}(\mathbb{A}, \mathbb{B}) = \log \left[ \frac{\mathcal{N}_{\mathbb{D}}(A_1, B_1)\mathcal{N}_{\mathbb{D}}(A_2, B_2)}{\mathcal{N}_{\mathbb{D}}(A_1, A_2)\mathcal{N}_{\mathbb{D}}(B_1, B_2)} \right]. \tag{2}$$

This inter-model prediction discrepancy will be close to 0 when the output discrepancy for models within the same set (numerator) is similar to that of models coming from different sets (denominator), and it will be greater than 0 when the discrepancy for models coming from different sets is greater than that of models coming from the same set. When applied to models trained on different demographic groups, we refer to this diverse set of models as a demographically-informed pool and $\Phi_{\mathbb{D}}$ becomes our *Demographically-Informed Prediction Discrepancy Index (DIPDI)*. This will be the case, for example, when models in $\mathbb{A}$ are trained on male individuals while models in $\mathbb{B}$ are trained on female individuals.

Note that in our current analysis, we have focused on model sets of size 2 for simplicity (e.g. both $\mathbb{A}$ and $\mathbb{B}$ have two elements). However, it is important to note that this concept can be extended to larger sets. The generalization involves considering combinations of pairs of models both within each set and between different sets. For an extension of DIPDI to handle groups with more than two models, please refer to Appendix A.1.

## 3.2 DIPDI as a proxy for anticipating bias issues

Our goal is to anticipate whether biases may arise with respect to a particular protected attribute $a$ in a novel dataset before annotated labels become available. Here we provide an example where the task at hand is age regression and the protected attribute $a$ indicates the gender of the individual, which for simplicity we take as *male* ($a = M$) or *female* ($a = F$). We create two sets of models (age regressors): $\mathbb{A}$, where models $A_i$ are trained only on male individuals, i.e. $a = M$; and $\mathbb{B}$, where models $B_i$ are trained only on female individuals, i.e. $a = F$. We say that this constitutes a demographically-informed pool of models, as each of them was trained on individuals from a particular demographic group characterized by the protected attribute $a$. Let us also have a fixed dataset $\mathbb{D}$ that will be used as the novel target population where potential biases would want to be flagged. $\mathbb{D}$ is a balanced dataset according to the protected attribute $a$ (but unlabeled with respect to output class, i.e. without the reference age). In our example, this means that $\mathbb{D}$ is composed of 50% male and 50% female individuals.

Our hypothesis is that for larger values of $\Phi_{\mathbb{D}}(\mathbb{A}, \mathbb{B})$, computed for a pool of models comprising sets $\mathbb{A}$ and $\mathbb{B}$, biases are more likely to emerge. In other words, we hypothesize that inconsistencies between the output discrepancy of models trained on highly imbalanced datasets with respect to the protected attribute $a$ will tend to co-occur with potential bias issues. To confirm our hypothesis, we first look for biases with respect to $a$ using ground-truth annotations in the target population (following a strategy similar to Larrazabal et al. (2020)), by computing performance gaps in terms of absolute error (using the ground-truth of each sub-population). Then we calculate DIPDI, which does not require ground-truth labels for $\mathbb{D}$, and verify if

it produces results that are in line with the conclusions we drew when using the annotations. As a sanity check, we incorporate control experiments where we break the assumption that sets $\mathbb{A}$ and $\mathbb{B}$ are trained on different demographic groups (e.g. by training models in $\mathbb{A}$ and $\mathbb{B}$ using balanced data formed by 50% male and 50% female individuals), and show that in these cases DIPDI returns values close to 0. Moreover, the theoretical derivation included in Appendix A.2 provides analytic expressions for the relationship between performance gap and output discrepancies captured by DIPDI, assuming one-dimensional Gaussian soft outputs for classification models.

## 4 Experimental validation

We start by verifying the behaviour of DIPDI under controlled conditions using synthetic data (Section 4.1). Then, we perform a set of experiments to evaluate the proneness to gender bias of tasks such as age estimation (from face and X-ray images) and younger vs. older classification of celebrities (from face images). Our focus is particularly on scenarios where a specific subgroup is underrepresented, as discussed in Larrazabal et al. (2020). We show that DIPDI anticipates potential biases against the minority group when training data is highly imbalanced in gender representation (Section 4.3). To this end, we employ ground-truth annotations for the target population to compute performance gaps for models trained with different imbalance ratios, in the different subgroups. Then, we proceed to compute DIPDI (which does not require ground-truth labels) in the target population. We show that bias gaps tend to occur for larger DIPDI values (Section 4.4). We conclude the study by showing how DIPDI can serve to anticipate potential bias issues at deployment in populations with distribution shifts, when target annotations are not yet available (Section 4.5).

### 4.1 DIPDI on synthetic data

We start by verifying the behaviour of the proposed DIPDI using synthetic data. The purpose is to show, using a simple example, how sensitive DIPDI is when measuring bias proneness. To this end, we generate a synthetic dataset composed of samples $\mathbf{s_i} = (\mathbf{x_i}, y_i)$, where $\mathbf{x_i} \in \mathbb{R}^2$ are bi-dimensional feature vectors coming from two bi-variate normal distributions $N_{a=0}(\mu_0 = (0,0), \Sigma_0 = I)$ and $N_{a=1}(\mu_1 = (1,1), \Sigma_1 = I)$. These distributions simulate different demographic groups characterized by a protected attribute $a = 1$ (e.g. female) or $a = 0$ (e.g. male). We then generate the corresponding labels $y_i \in \mathbb{R}$ for each data sample to simulate a regression problem (e.g. age regression). We do this by assigning $y_i$ to be equal to the first feature dimension of the sample, linearly interpolated to ensure it is within 1 and a 100 years, with an added uniform random noise of 10 years, simulating an age estimation scenario (see Figure 7 in the Appendix B.1). We then train support vector regression (SVR) models on these samples, perform inference, and subsequently compute DIPDI based on the actual predictions produced by these models.

Importantly, to ensure that under-representation of a certain group will bias age regression models to exhibit better performance in the majority group, we sample each distribution by varying the proportion of male and female samples in training. The range of gender imbalance cases spans from 100-0 (100% male) to 0-100 (100% female), with increments of 10%. We generated 10,000 training samples that were partitioned into 10 folds for statistical purposes, and an additional 1,000 samples for testing that were held constant across all experiments.

The results of the synthetic experiment are presented in Figure 1. At the top (Figure 1a), the mean absolute error (MAE) is shown for models trained with different imbalance ratios evaluated on males, females, and the whole population. We can see performance disparities across demographic groups, which highlights the impact of gender imbalance on predicted accuracy. The corresponding DIPDI values are shown in Figure 1b. The observed results are consistent with our hypothesis that DIPDI helps to anticipate bias proneness. In Appendix B.2 we include results for an even simpler synthetic experiment, where instead of simulating the dataset and training SVR models, we directly simulate the model outputs and show how DIPDI reacts in highly controlled conditions.

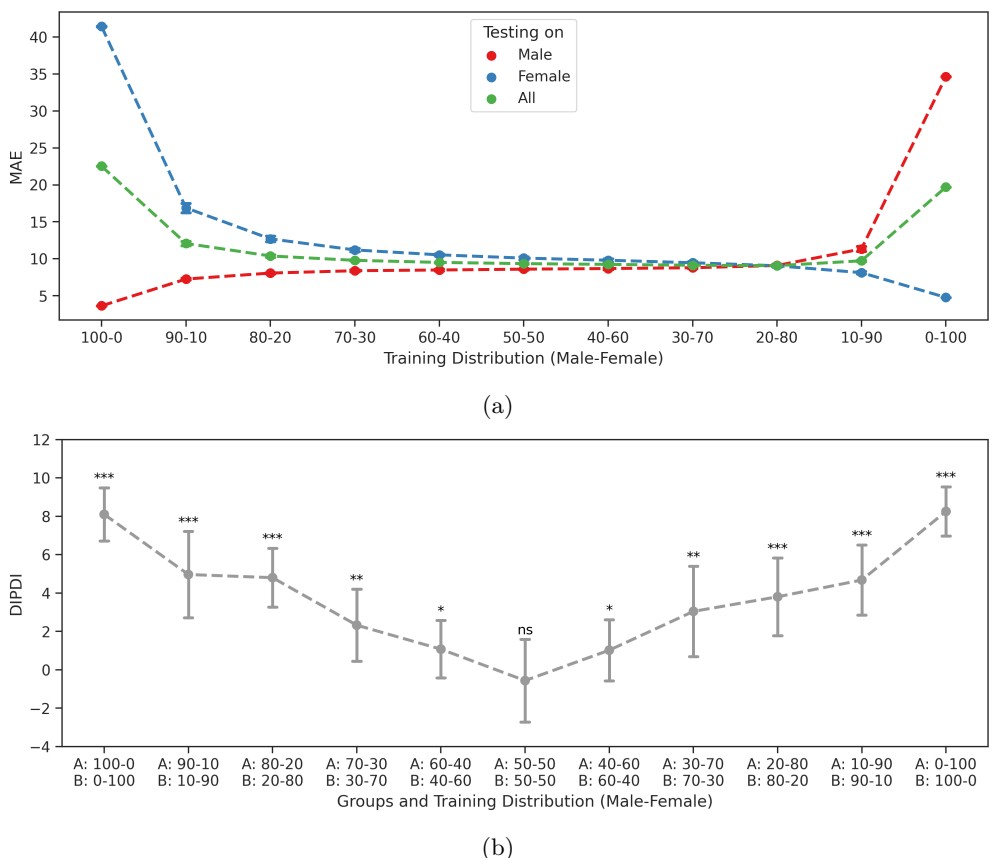

(a)

(b)

Figure 1: (a) Mean absolute error (MAE) of SVR models trained on synthetic datasets of age estimation with varying gender imbalance ratios. (b) DIPDI values for different group compositions, highlighting the impact of gender imbalance in prediction consistency. Statistical significance with respect to a mean equal to 0 was measured by a Wilcoxon test (ns: non-significant, *: p-value $< 0.05$, **: p-value $< 0.01$, ***: p-value $< 0.001$).

### 4.2 DIPDI on real scenarios: datasets and experimental setup

We conduct experiments on the task of age regression using convolutional neural networks (CNNs), employing three public databases: ChestX-ray14 (Wang et al., 2017), UTKFace (Zhang et al., 2017) and IMDB-WIKI (Rothe et al., 2018). We also evaluate DIPDI for classification problems, considering a binary classification (younger vs older celebrities) using the CelebA dataset (Liu et al., 2015). All experiments were performed using PyTorch (Paszke et al., 2017) and executed on an NVIDIA Titan X GPU.[1].

**ChestX-ray14 dataset.** The ChestX-ray14 dataset contains 112,120 high-resolution frontal-view radiographs of 30,805 unique patients with age and gender labels. Each image is annotated with up to 14 different chest disease labels extracted from radiology reports (not used in our study), age and gender. We use the ChestX-ray14 dataset to perform subgroup analysis in terms of gender and to evaluate DIPDI in models trained to perform age estimation from radiological images. We use here for gender the binary labels reported in the dataset, i.e. male and female. To avoid having two images of the same patient in train and test, we randomly selected one image per patient, resulting in a total of 28,350 images which include healthy and pathological cases. This database was divided into 10 folds using a stratified cross-validation strategy, where each fold is balanced by gender. For each cross validation instance, one fold is used to evaluate the model and the remaining 9 folds are used to train the model, which are further sub-divided into training (90%) and validation (10%) subsets for hyper-parameter tuning and model selection.

For all experiments on ChestX-ray14 we used a DenseNet-121 (Huang et al., 2017) pretrained on ImageNet (Russakovsky et al., 2015). The last layer of the network was replaced with an adaptive pooling layer, followed by a single-output neuron layer to predict age. The models were trained for 50 epochs using the Adam optimizer (Kingma & Ba, 2014) with default parameters and the mean absolute error (MAE) loss function.

**UTKFace dataset.** The UTKFace dataset is a collection of over 20,000 facial images spanning ages from 0 to 116, annotated for age, gender, and ethnicity. It exhibits diverse variations in pose, facial expression, lighting, occlusion, and resolution. Images were filtered to include ages from 10 to 100 and followed the same training settings as applied in the case of ChestX-ray14. This dataset is utilized for subgroup analysis and assessing DIPDI in age estimation models.

We employed a VGG-16 architecture (Simonyan & Zisserman, 2014), pretrained on ImageNet, with the final layer replaced by adaptive pooling and a single-output neuron layer for age prediction.

**IMDB-WIKI dataset.** The IMDB-WIKI dataset consists of 523,051 face images of 20,284 celebrities collected from IMDB and Wikipedia with age and gender labels. Age is estimated from the date of birth and the year when the photo was taken. The IMBD-WIKI dataset is used to perform subgroup analysis and to evaluate DIPDI for models trained to perform age estimation from facial images.

We used a VGG-19 architecture (Simonyan & Zisserman, 2014) pre-trained on ImageNet. We added a single-output neuron layer with ReLU activation and fine-tuned the last four layers. The models were trained with a MAE loss for 10 epochs using the Adam optimizer with default parameters.

**CelebA dataset.** The CelebFaces Attributes (CelebA) dataset (Liu et al., 2015) is a large-scale repository comprising over 200,000 celebrity images, each annotated with 40 attributes. This dataset covers diverse facial poses and background variations. In our study, we choose the 'young' attribute as the target label (thus classifying younger vs older individuals) for prediction, and balance both gender and target to mitigate potential spurious correlations.

To conduct subgroup analysis and evaluate DIPDI in classification models, we employed a ResNet-50 architecture (He et al., 2016) pretrained on ImageNet, replacing the final layer with a two-output neuron layer for binary classification. The training process followed the same protocols as ChestX-ray14 and UTKFace, employing the cross-entropy loss.

---

[1]Our code is publicly available at `https://github.com/lamansilla/DIPDI-Biases`

### 4.3 Assessing the impact of gender imbalance in a supervised setting

**Age estimation from X-ray images.** We analyze the impact of gender imbalance in age estimation from radiological images by performing a supervised subgroup evaluation. The aim is to understand if the age estimation task is prone to be biased with respect to gender if a certain subgroup is under-represented. We will then see if the proposed DIPDI can predict such behaviour without ground-truth annotations. We train models with different degrees of gender imbalance and then examine their performance separately in male and female subgroups. We consider five cases of gender imbalance in training: 100-0, 75-25, 50-50, 25-75, and 0-100. Importantly, male and female subgroups in the test population are always equal in size. This means that every model is evaluated on equal footing.

The MAE for ChestX-ray14 is shown in Figure 2a. The results show that imbalance with respect to the protected attribute leads to a significant difference in performance across subgroups, confirming that this problem is prone to be biased with respect to gender if there is under-representation in the training dataset. For example, when testing on female subjects, models trained only on male (100-0) data have higher MAE than models trained on female images. The same happens when testing on female individuals: models trained only on female data (0-100) significantly outperform those trained on male data. Moreover, the differences between male and female subgroups are less significant when the training data is less imbalanced. These results are consistent with previous observations reported by Larrazabal et al. (2020) in the context of disease prediction from X-ray images. Appendix B.3 contains supplementary results for ChestX-ray14 including additional statistics and metrics.

**Age estimation from face images.** We also perform a similar analysis to study the impact of gender imbalance in age estimation from facial images. For UTKFace, we followed the same procedure as for ChestX-ray14. Figure 2c shows the MAE results for models trained with varying degrees of gender imbalance and tested on male and female subgroups over 10 folds. These results demonstrate a significant performance disparity across subgroups resulting from an imbalance in the protected attribute.

In the case of IMDB-WIKI, we train an ensemble of 5 models with different degrees of male-female imbalance and then evaluate their performance separately in male and female subgroups. We perform 20-fold cross-validation with a 60/20/20 ratio for the training, validation, and test sets.

The MAE is shown in Figure 2e for these experiments. We observe that the models perform better in the subgroup (either male or female) that is most represented in training, while their performance deteriorates in the other subgroup.

**Younger vs older classification from face images.** We explore the impact of gender imbalance when classifying celebrity photos using the CelebA dataset as either younger vs older individuals. Figure 2g presents the accuracy results by subgroup over 10 folds for models trained with varying gender imbalance ratios. The results reveal large gaps in performance for different demographic subgroups in highly imbalanced cases, while the gaps reduce for more balanced cases. We include supplementary results for CelebA in Appendix B.4, which include confusion matrices and a fairness metric (Equality of Opportunity).

### 4.4 Estimating bias proneness without ground-truth via DIPDI

In the previous section we confirmed that age estimation and classification of younger vs older individuals are tasks prone to be biased with respect to gender, by computing error gaps between subgroups using ground-truth annotations. Now, we want to study if it is possible to measure such bias proneness in a given population without ground-truth labels using DIPDI. We are interested in analyzing if output discrepancy for demography-aware model sets can be used as a proxy for anticipating potential fairness problems in specific ML tasks. To this end, we compute DIPDI in the same settings where we explicitly evaluated biases in the previous section. Note that we choose different discrepancy functions $d(\cdot, \cdot)$ (from Equation 1) for regression and classification problems. For regression problems, we choose $d$ to be the absolute error between the model predictions. For classification, as we output soft probabilities, we adopt the Jensen–Shannon divergence between the output distributions.

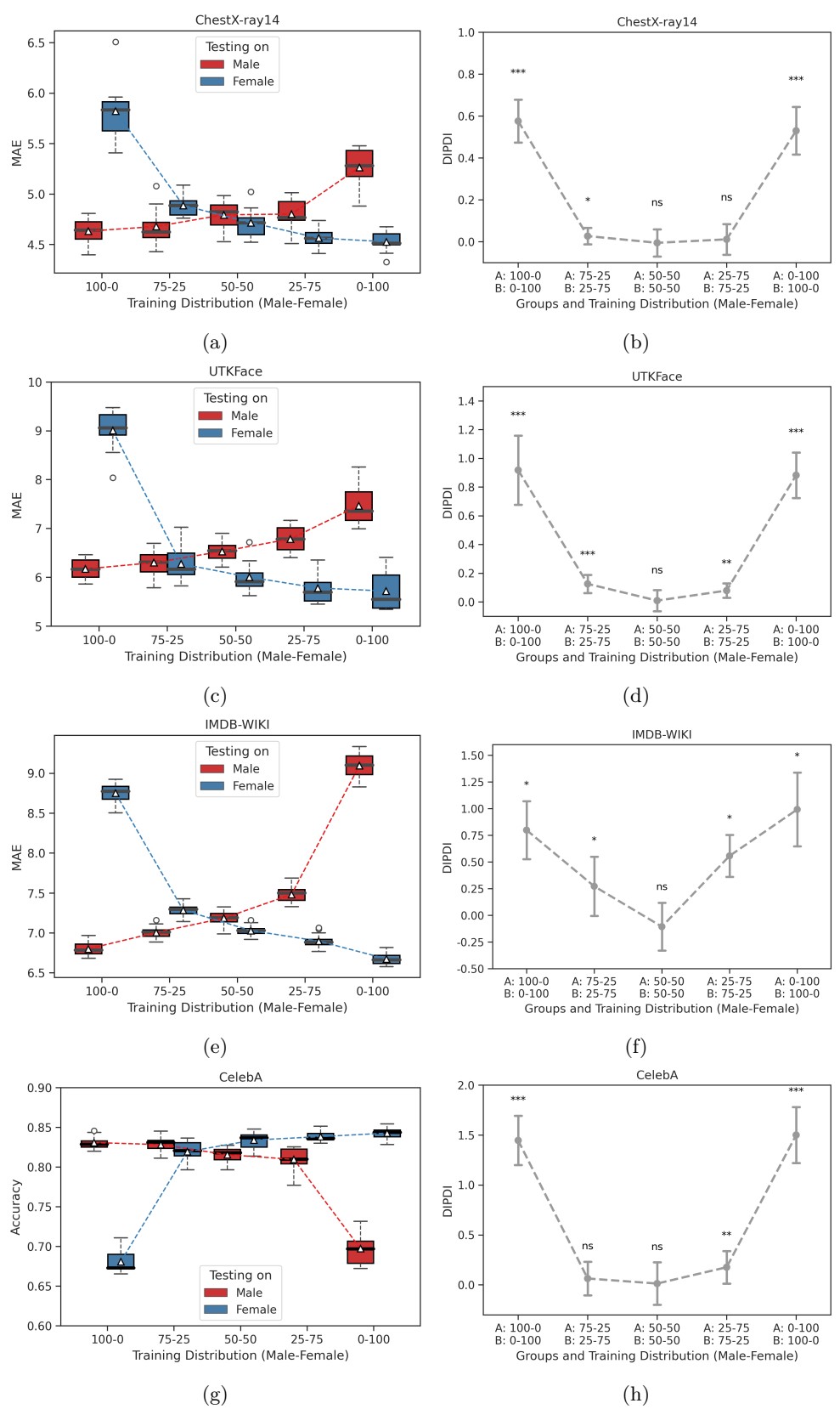

Figure 2: Results for age estimation (rows 1-3) and younger vs older classification (row 4) across models trained with different gender imbalance ratios. Model performance, measured through MAE and Accuracy, shows wider gaps in highly imbalanced cases. It is effectively captured by DIPDI with higher values for imbalance, and close-to-zero values for balanced cases.

In all datasets, we consider different scenarios of gender imbalance for the set of models $\mathbb{A}$ and $\mathbb{B}$ to be evaluated. Five comparisons are made: 100-0 vs 0-100, 75-25 vs 25-75, 50-50 vs 50-50, 25-75 vs 75-25, and 0-100 vs 100-0. To control for finite-size sampling variability, we split the training data into four random disjoint partitions, so that no data is shared between models even when they are trained for the same demographic sub-group (i.e. even though $A_1$ and $A_2$ are trained on subjects from the same demographic group, the exact individuals are not the same). Then DIPDI is computed on the held-out test set (i.e. the unlabeled data $\mathbb{D}$), which is balanced by gender. Additional results for DIPDI are included in Appendix B.5.

The plots in the right column of Figure 2 show DIPDI for the age regression and younger vs older classification tasks. Note that, in all scenarios, the index values are very close to 0 when comparing sets of models trained in the same population, but higher than 0 when comparing models from different populations, in line with the absence or presence of biases as a function of data imbalance shown in the corresponding left column. Taken together these results demonstrate the co-occurrence between higher DIPDI and bias proneness: models coming from the same demographic population, produce more consistent outputs when evaluated on a target population. This output stability is clearly evidenced by index values close to 0 for 50-50 distributions. In contrast, the index returns significantly higher values when it comes to models trained with different demographic subgroups, where biases are in turn prone to appear, as shown in our previous supervised analysis (Section 4.3). Importantly, note that no labels were required in the target population $\mathbb{D}$ when computing DIPDI.

## 4.5 Anticipating potential demographic biases in distribution shift scenarios with DIPDI

We have highlighted in the previous section the role of DIPDI in identifying potential demographic biases in populations that lack ground-truth annotations. Now, we turn our attention to a new challenge: demonstrating how DIPDI can deal with domain shift scenarios even when ground-truth data is unavailable. Prior research has identified the vulnerability of fairness properties of machine learning models when deployed on datasets differing from those used during model development (Schrouff et al., 2022). In this context, we leverage DIPDI as an unsupervised alternative to traditional fairness metrics for understanding bias proneness in populations under different types of distribution shifts. Here we focus on two cases: covariate shift, where the conditional distribution of the input features (e.g. pixel intensities) changes between source and target population; and label shift, where the conditional distribution of labels change between source and target (e.g. different prevalence for a disease, or different age distributions in our age regression problem). As discussed in Schrouff et al. (2022), such shifts as well as other types of changes in data distribution, may result in failures of fairness transfer across distribution shifts. In other words, models that were not biased in a source distribution may start to exhibit biases in the target distribution. To this end, we explore two different scenarios.

### 4.5.1 Label shift: age distribution experiment

This experiment involves a target population that is always balanced by gender, and we introduce label shifts by altering the age distribution within one gender group (either male or female, but not both). Specifically, we increase the proportion of individuals with ages exceeding a predefined limit (set at 45 in our experiments), while maintaining the age distribution within the non-shifted group. For each shift scenario considered, we calculate both the DIPDI and the MAE gap ($\Delta$MAE) between male and female models, when tested separately on male and female subsets. For the male subset, we calculate the $\Delta$MAE by subtracting the MAE of a female-trained model from that of a male-trained model. Similarly, for the female subset, we subtract the MAE of a male-trained model from that of a female-trained model. Note that the calculation of $\Delta$MAE requires access to ground truth annotations, whereas DIPDI does not.

Figure 3 presents the mean and standard deviation of DIPDI and $\Delta$MAE for age shift ratios ranging from 50% to 90% (a shift ratio of 90%, for example, implies 90% of the subpopulation is under 45, and 10% is at or above 45). Our aim here is to understand if such label shift results in a task that is more prone to be biased with respect to the demographic groups under analysis. In principle, there could be three possibilities when compared with the original distribution: the problem is more, equally or less prone to be biased. Note that when the shift affects the male group (Fig. 3a), DIPDI tends to slightly increase, and a corresponding

slightly increasing gap is observed for both male and female test groups. On the other hand, when varying the proportion of females younger than 45 years old in the unseen population, we observe that the $\Delta$MAE between models trained on male and female individuals stays constant for males (red curve in Fig. 3b, right panel), but decreases for female subjects (blue curve), reaching a level of bias proneness equivalent to the one observed for males (at 90%, where the blue and red curves intersect). In other words, decreasing the age of the female population changes the bias proneness of that group (as measured by analyzing the $\Delta$MAEs), making it more fair as it reaches levels of bias proneness equivalent in both populations. As expected, the DIPDI index follows exactly the same tendency (is reduced as bias proneness is reduced), confirming our hypothesis. This different behavior observed when introducing label shifts in the test male and female groups may be rooted in the different ways in which the features of each group interact with age. In fact, for this dataset it is well known that the baseline performance is different for both groups. This could explain the greater susceptibility of a group to changes in age distribution.

These experiments underscore the effectiveness of DIPDI in a label shift scenario, particularly when ground-truth annotations are unavailable. An increase in DIPDI during deployment, compared to the development phase, can be interpreted as an indicator of intensifying bias proneness within one or both demographic groups, whereas a decrease in DIPDI implies a potential reduction in bias within these groups.

### 4.5.2 Covariate shift: color distribution experiment

In this experiment, we evaluate a classification model trained for a new task: distinguishing between blond and non-blond celebrities in the CelebA dataset. While the model is trained on the original color images, we induce a covariate shift in the target population by transforming color test images (RGB) to grayscale. We guarantee target and gender balance in both training and test splits to avoid spurious correlations in our experiment, executing 10 runs with different random seeds.

Figure 4 presents DIPDI values and the accuracy gap ($\Delta$Acc) between models trained on male and female subjects, tested on the whole target population with covariate shift (grayscale images) and without covariate shift (RGB images). We note that evaluating bias proneness of the original models in the shifted distribution using the ground-truth annotations shows decreased bias overall. Notably, the $\Delta$Acc between models trained on male and female subjects becomes more similar for both groups for grayscale images compared to RGB. The DIPDI index consistently captures this behavior, resulting in a decrease when computed in grayscale images.

## 5 Discussion

In this work, we tackle the issue of anticipating potential demographic biases at deployment in the absence of ground truth annotations. Typical methods designed to assess fairness require access to such annotations, which may be available at training time but not when deploying models for previously unseen data. A prototypical example of this would be a model trained on a public dataset which will then be applied to a local population for which we do not have the corresponding annotations. Recent work has highlighted how distribution shifts may affect fairness (Schrouff et al., 2022), resulting in a potential risk. While an explicit fairness metric may not be computed, we argue that we can employ the discrepancy of output predictions from pools of models trained on different demographic groups as a proxy to provide an early warning about potential demographic biases. We propose a concrete solution in terms of an index, DIPDI, whose value indeed provides a measure for the proneness towards biased solutions.

Intuitively, we can think about output discrepancies in a set of models as a notion of uncertainty, which in ensemble models is usually estimated as the variance in the predictions of components of the ensemble (Lakshminarayanan et al., 2017). If all models in an ensemble agree on a prediction, the uncertainty for this sample is likely low. Conversely, if there is a high variance in the predictions across the models, this indicates higher uncertainty. When we evaluate the discrepancy in the predictions of models trained for a particular demographic group, we could interpret them as models within an ensemble, and consequently the discrepancy in their predictions for a given subject could be seen as the uncertainty of the ensemble. In that sense, our index quantifies the relative uncertainty estimated when using 'ensembles' of models trained with

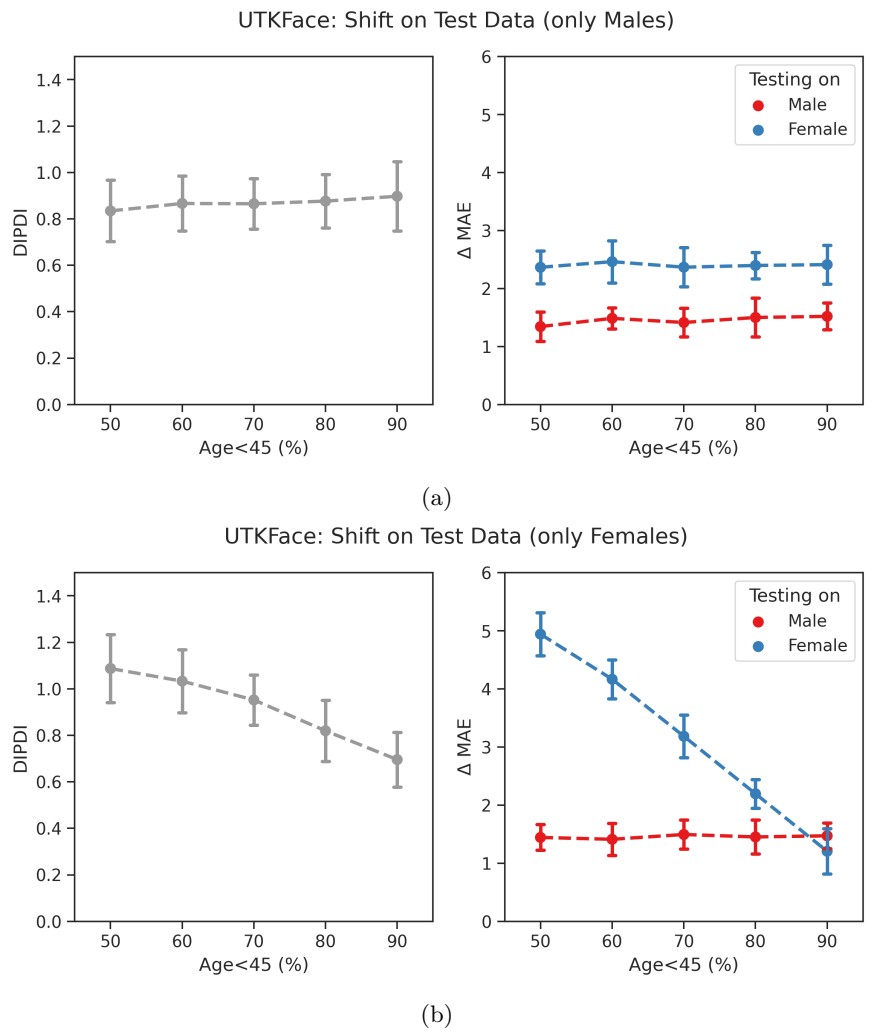

(a)

(b)

Figure 3: Mean and standard deviation DIPDI (left) and MAE gap (right) under label shift scenarios for male (a) and female (b) groups separately on UTKFace. Label shift is induced by modifying the age distribution of individuals under 45 years at increasing ratios for male and female test groups independently. Note that in both cases of shift, the DIPDI tends to follow the behaviour of the difference curve, showing an increase with increasing biases and a decrease with decreasing biases.

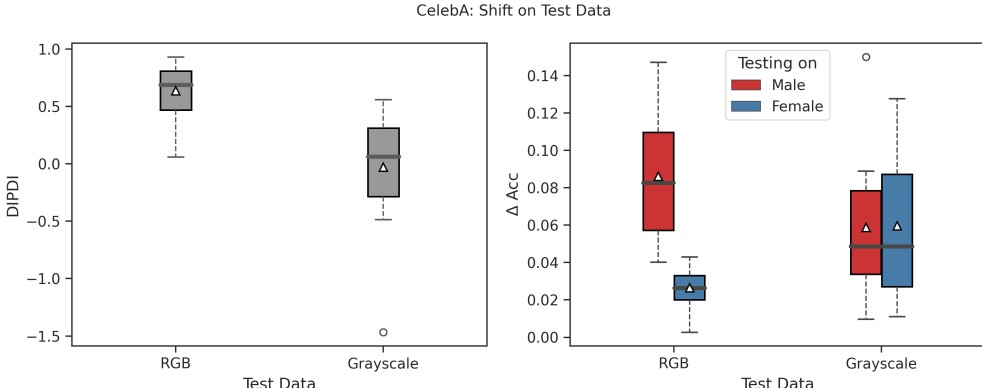

Figure 4: DIPDI values (left) and accuracy gap (right) corresponding to covariate shift (RGB to grayscale) in CelebA for blond classification. Note that accuracy gaps become more similar as images transform from RGB to grayscale, supported by a decrease in DIPDI.

data from different demographic groups (numerator) and from the same demographic group (denominator). If both are similar (ratio equal to 1), then we get a DIPDI value close to 0 (log ratio 1) indicating that the problem shows no early signs of potential bias with respect to the analyzed demographic values. However, higher discrepancies (uncertainty) for models from different demographic groups will lead to DIPDI values significantly larger than 0, indicating bias proneness for the task under analysis. In particular, an increase in DIPDI from model development (training) to deployment could be interpreted as a red flag, triggering further detailed assessment. We showed that DIPDI can also be used to understand how fairness transfers across distributions, validating our assumption in scenarios involving label and covariate distribution shifts. Other types of distribution shifts (Quinonero-Candela et al., 2008), and even compound shifts as discussed in Schrouff et al. (2022) could also be considered, but would require further validation.

We note that while we have expressed DIPDI here as a global population average, the same reasoning could in principle be applied to population subsets defined by the intersection of multiple demographic traits (i.e. intersectional fairness), or even on a subject-by-subject basis, closer to the definition of individual fairness. Such predictive discrepancies as captured by DIPDI could serve to flag subjects or sub-groups at higher risk of suffering biases, constituting another avenue of research to explore in future work. Regarding counterfactual fairness approaches, DIPDI shares some resemblance as it involves a form of hypothetical scenario analysis. However, DIPDI's approach is more empirical, focusing on the discrepancies in predictions of actual models trained on different populations, while counterfactual fairness measures often involve more complex causal modeling and assumptions. Finally, in relation to approaches based on fairness through unawareness that simply exclude protected attributes from the model, DIPDI actively measures the impact of these attributes by training separate models on different demographic groups. Overall, we believe that DIPDI offers a fresh perspective in the fairness literature, focusing on the unsupervised setting, which is not commonly discussed in this field, and may spark new discussions towards developing novel unsupervised bias discovery methods to anticipate bias issues in the absence of ground truth.

**Acknowledgments**

This work was supported by Argentina's National Scientific and Technical Research Council (CONICET), which covered the salaries of R.E., D.H.M. and E.F., as well as the fellowships of L.M. and E.C. The authors gratefully acknowledge NVIDIA Corporation for provinding GPU computing, the support of Universidad Nacional del Litoral (Grants CAID-PIC-50220140100084LI, 50620190100145LI), Agencia Nacional de Promoción de la Investigación, el Desarrollo Tecnológico y la Innovación (Grants PICT 2018-3907, PRH 2017-0003, PICT-2020-SERIEA-01765, PRH 2022-00002) and the Google Award for Inclusion Research (AIR) Program.

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

# A DIPDI: General formulation and theoretical analysis

## A.1 General formulation of DIPDI

In the main manuscript (Section 3.1) we introduced DIPDI for pools of 2 models for simplicity. However, having more models in each pool could help to reduce noise in the estimation. Thus, here we introduce a more general formulation for pools $\mathbb{A} = \{A_1, ..., A_m\}$ and $\mathbb{B} = \{B_1, ..., B_m\}$ of $m$ models each, with $m$ denoting an even integer. The generalized formulation can be expressed as

$$\Phi_{\mathbb{D}}(\mathbb{A}, \mathbb{B}) = \frac{1}{\log_2 m} \log \left[ \frac{\prod_{i=1}^{m} \mathcal{N}_{\mathbb{D}}(A_i, B_i)}{\prod_{i=1}^{m/2} \mathcal{N}_{\mathbb{D}}(A_i, A_{m/2+i}) \mathcal{N}_{\mathbb{D}}(B_i, B_{m/2+i})} \right]. \tag{3}$$

To illustrate, let us exemplify the case when $m = 4$:

$$\Phi_{\mathbb{D}}(\mathbb{A}, \mathbb{B}) = \frac{1}{2} \log \left[ \frac{\mathcal{N}_{\mathbb{D}}(A_1, B_1) \mathcal{N}_{\mathbb{D}}(A_2, B_2) \mathcal{N}_{\mathbb{D}}(A_3, B_3) \mathcal{N}_{\mathbb{D}}(A_4, B_4)}{\mathcal{N}_{\mathbb{D}}(A_1, A_3) \mathcal{N}_{\mathbb{D}}(A_2, A_4) \mathcal{N}_{\mathbb{D}}(B_1, B_3) \mathcal{N}_{\mathbb{D}}(B_2, B_4)} \right]. \tag{4}$$

By rearranging factors, we obtain

$$\Phi_{\mathbb{D}}(A, B) = \frac{1}{2} \log \left[ \frac{\mathcal{N}_{\mathbb{D}}(A_1, B_1) \mathcal{N}_{\mathbb{D}}(A_3, B_3)}{\mathcal{N}_{\mathbb{D}}(A_1, A_3) \mathcal{N}_{\mathbb{D}}(B_1, B_3)} \right] + \frac{1}{2} \log \left[ \frac{\mathcal{N}_{\mathbb{D}}(A_2, B_2) \mathcal{N}_{\mathbb{D}}(A_4, B_4)}{\mathcal{N}_{\mathbb{D}}(A_2, A_4) \mathcal{N}_{\mathbb{D}}(B_2, B_4)} \right] \tag{5}$$

$$\Phi_{\mathbb{D}}(A, B) = \frac{1}{2} \big( \Phi_{\mathbb{D}}(\{A_1, A_3\}, \{B_1, B_3\}) + \Phi_{\mathbb{D}}(\{A_2, A_4\}, \{B_2, B_4\}) \big) \tag{6}$$

In other words, our generalized formulation enables computing DIPDI for sets of $m$ models, by considering it as the average of pairwise DIPDIs for different subsets of size 2.

To better understand the definition of the products in Eq. 3, we can imagine an $m \times m$ matrix where rows and columns represent models in groups $\mathbb{A}$ and $\mathbb{B}$, respectively. For the products in the numerator, we simply take the main diagonal of the matrix. For the products in the denominator, we focus on the upper diagonal of $m/2$ elements. A visual representation of this idea is presented in Table 1, illustrating the scenario for the specific case where $m = 4$.

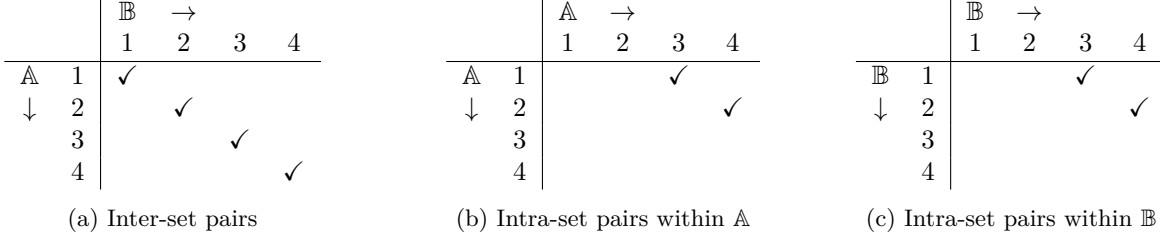

(a) Inter-set pairs          (b) Intra-set pairs within $\mathbb{A}$          (c) Intra-set pairs within $\mathbb{B}$

Table 1: Illustration of product definitions for DIPDI when $m = 4$. The ✓denotes the presence of a pair of models in factors $\mathcal{N}_{\mathbb{D}}(\cdot, \cdot)$ of Eq. 5.

## A.2 Theoretical analysis of the relationship between DIPDI and performance gap

In order to generate an intuition behind why DIPDI may anticipate biases, we will resort to a theoretical analysis in a simplified scenario. We will work with a binary classification problem, where the soft scores of the models will be taken to be one dimensional and assumed normally distributed. We call $X_{\mathbb{A}}$ and $X_{\mathbb{B}}$ the distributions of outputs for models from set $\mathbb{A}$ and $\mathbb{B}$ respectively. For simplicity, we will work with the outputs corresponding to the positive target class, which we take to correspond to values on the left of the decision boundary, but the problem is symmetrical and the same derivation can be replicated for the

negative class. In Fig. 5a we illustrate this scenario with two model sets (in blue and green) that produce different error rates, given by their corresponding shaded areas on the other side of the decision boundary. Since $X_{\mathbb{A}}$ and $X_{\mathbb{B}}$ are normally distributed, we characterize them by their respective means ($\mu_{\mathbb{A}}$ and $\mu_{\mathbb{B}}$) and standard deviations ($\sigma_{\mathbb{A}}$ and $\sigma_{\mathbb{B}}$). The error gap between these sets of models for a given decision boundary

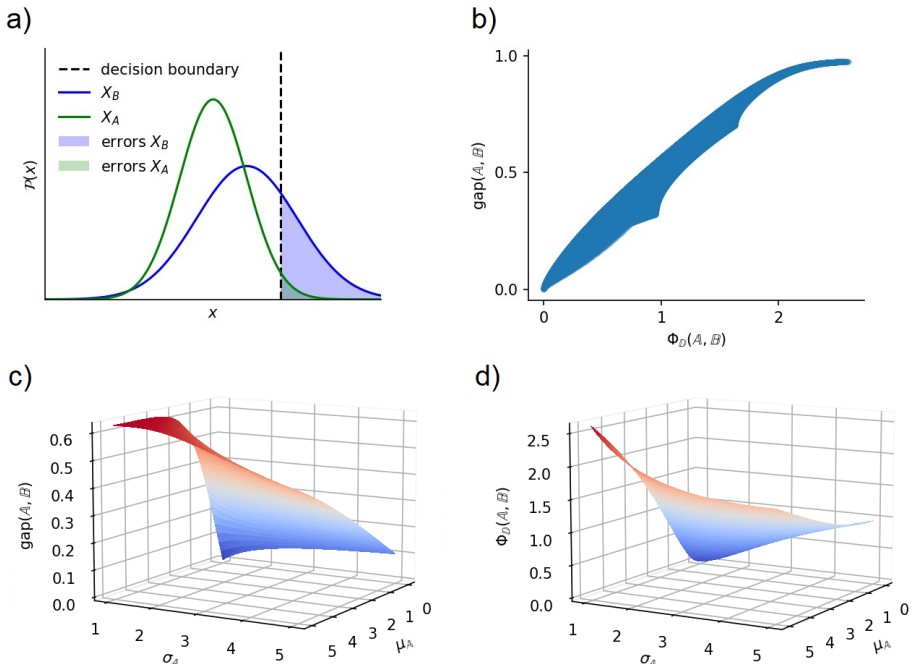

Figure 5: a) Sketch of the distributions of soft outputs for two sets of models $\mathbb{A}$ (in green) and $\mathbb{B}$ (in blue). The respective error rates correspond to the fraction of outputs beyond the decision boundary (dashed line), indicated as shaded regions of the same color. b) Scatter plot of the error gap vs. DIPDI for multiple combinations of distribution parameters $\mu_{\mathbb{B}}$ and $\sigma_{\mathbb{B}}$, for fixed $\mu_{\mathbb{A}}$ and $\sigma_{\mathbb{A}}$. c) & d) Surface plots for the error gap and DIPDI, respectively, for the same parameter configurations as in panel b).

$b$ can then be written by integration as

$$\mathrm{gap}(\mathbb{A}, \mathbb{B}) = \left| \int_b^\infty \mathcal{N}\left(x\,;\mu_{\mathbb{B}}, \sigma_{\mathbb{B}}\right)\,dx - \int_b^\infty \mathcal{N}\left(x\,;\mu_{\mathbb{A}}, \sigma_{\mathbb{A}}\right)\,dx \right|. \tag{7}$$

DIPDI is a measure of the mean discrepancy between outputs of two model sets, relative to the mean discrepancies of outputs within sets. Without loss of generality, we can take the squared error as the discrepancy function to simplify DIPDI in this analytical interpretation. Since we assume the model outputs are normally distributed, the mean squared distance between outputs is given by

$$\mathcal{E}\left[d_{\mathbb{A}\mathbb{B}}^2\right] = \mu_{\mathbb{A}\mathbb{B}}^2 + \sigma_{\mathbb{A}\mathbb{B}}^2, \quad \text{with} \tag{8}$$

$$\mu_{\mathbb{A}\mathbb{B}} = \mu_{\mathbb{A}} - \mu_{\mathbb{B}} \quad \text{and} \tag{9}$$

$$\sigma_{\mathbb{A}\mathbb{B}}^2 = \sigma_{\mathbb{A}}^2 + \sigma_{\mathbb{B}}^2. \tag{10}$$

For points from within the same distribution, we have in turn

$$\mathcal{E}\left[d_{\mathbb{A}\mathbb{A}}^2\right] = 2\sigma_{\mathbb{A}\mathbb{A}}^2, \tag{11}$$

$$\mathcal{E}\left[d_{\mathbb{B}\mathbb{B}}^2\right] = 2\sigma_{\mathbb{B}\mathbb{B}}^2. \tag{12}$$

So that in the limit of considering a large number of pairs of models, and for $\mathcal{N}_\mathbb{D} = \sqrt{\mathcal{E}[d^2(\cdot,\cdot)]}$, the DIPDI becomes

$$\Phi_\mathbb{D}(\mathbb{A},\mathbb{B}) = \log\left[\frac{(\mu_\mathbb{A} - \mu_\mathbb{B})^2 + \sigma_\mathbb{A}^2 + \sigma_\mathbb{B}^2}{2\,\sigma_\mathbb{A}\,\sigma_\mathbb{B}}\right]. \tag{13}$$

Note that DIPDI is then sensitive to both a difference in the mean and in the variance between the model predictions from two sets. In Fig. 5b-d, we have kept the parameters of $\mathbb{A}$ fixed ($\mu_\mathbb{A} = 0$, $\sigma_\mathbb{A} = 1$) while varying those of $\mathbb{B}$. The discrepancy grows if the distributions have different means, and also if the variances are different. Intuitively, if the mean is closer or further away from the decision boundary, the error rate will change. In that case the error gap is due to a systematic shift in the predictions. In turn, if the variances are different, then the reliability of the two model sets are different, and DIPDI can also sense that. Indeed we observe that higher DIPDI values correspond to higher error gaps Fig. 5b. In what follows we provide analytic expressions for the relationship between performance gap and output discrepancies when either the means or the variances of both sets are different.

### A.2.1 DIPDI and unreliability

We first study the case where $\mu_\mathbb{A} = \mu_\mathbb{B}$. Without loss of generality we take $\sigma_\mathbb{A}$ fixed and let $\sigma_\mathbb{B}$ vary (see Fig. 6a). In this case we have

$$\mathcal{E}\left[d_{\mathbb{A}\mathbb{B}}^2\right] = \mu_{\mathbb{A}\mathbb{B}}^2 + \sigma_{AB}^2 = \sigma_\mathbb{A}^2 + \sigma_\mathbb{B}^2 \tag{14}$$

$$\Phi_\mathbb{D}(\mathbb{A},\mathbb{B}) = \log\left[\frac{\sigma_\mathbb{A}^2 + \sigma_\mathbb{B}^2}{2\,\sigma_\mathbb{A}^2}\right]. \tag{15}$$

Solving for $\sigma_\mathbb{B}$, we can compute the gap as

$$\mathrm{gap}(\mathbb{A},\mathbb{B}) = \left|\int_b^\infty \mathcal{N}\left(x\,;\mu_\mathbb{B},\sqrt{\mathcal{E}\left[\delta_{\mathbb{A}\mathbb{B}}^2\right] - \sigma_\mathbb{A}^2}\right)\,dx - \int_b^\infty \mathcal{N}\left(x\,;\mu_\mathbb{A},\sigma_\mathbb{A}\right)\,dx\right|. \tag{16}$$

We see from this equation that, as long as $\mu_\mathbb{B} < b$ so that the mean of $X_\mathbb{B}$ is on the correct side of the boundary, the gap is a monotonically increasing function of the mean discrepancy. Indeed, we can see how the gap increases with DIPDI in Fig. 6b.

### A.2.2 DIPDI and systematic errors

We then study the case where $\sigma_\mathbb{A} = \sigma_\mathbb{B}$. Again, without loss of generality we take $\mu_\mathbb{A} = 0$ fixed and let $\mu_\mathbb{B}$ vary (Fig. 6c). In this case we have

$$\mathcal{E}\left[d_{\mathbb{A}\mathbb{B}}^2\right] = \mu_{\mathbb{A}\mathbb{B}}^2 + \sigma_{\mathbb{A}\mathbb{B}}^2 = \mu_\mathbb{B}^2 + 2\,\sigma_\mathbb{A}^2 \tag{17}$$

$$\Phi_\mathbb{D}(\mathbb{A},\mathbb{B}) = \log\left[\frac{\mu_\mathbb{B}^2 + 2\sigma_\mathbb{A}^2}{2\,\sigma_\mathbb{A}^2}\right]. \tag{18}$$

Solving for $\mu_\mathbb{B}$, we can compute the gap as

$$\mathrm{gap}(\mathbb{A},\mathbb{B}) = \left|\int_b^\infty \mathcal{N}\left(x\,;\sqrt{\mathcal{E}\left[d_{\mathbb{A}\mathbb{B}}^2\right] - 2\,\sigma_\mathbb{A}^2},\sigma_\mathbb{B}\right)\,dx - \int_b^\infty \mathcal{N}\left(x\,;\mu_\mathbb{A},\sigma_\mathbb{A}\right)\,dx\right| \tag{19}$$

Once again, we see from this equation that the gap is a monotonically increasing function of the mean discrepancy, and a tight correlation between DIPDI and gap is present (see Fig. 6d).

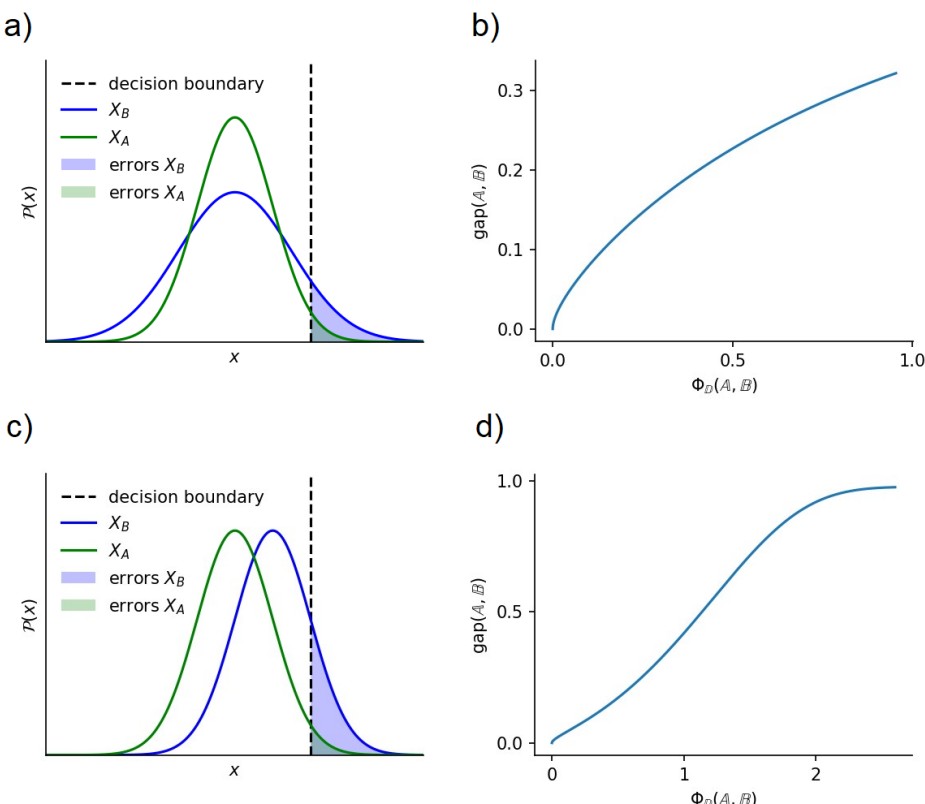

Figure 6: a) & c) Sketch of the distributions of soft outputs for two sets of models $\mathbb{A}$ (in green) and $\mathbb{B}$ (in blue). The respective error rates correspond to the fraction of outputs beyond the decision boundary (dashed line), indicated as a shaded regions of the same color. In a) the models have the same mean but different variance, while in c) the models have the same variance but different mean. b) & d) Error gap vs DIPDI for models with either different variance (b) or different means (d).

## B  Experimental validation: Additional results

### B.1  DIPDI on synthetic data: simulating distributions

Figure 7 shows examples of datasets simulated for the synthetic experiment in Section 4.1 of the main manuscript. These datasets are generated by varying the proportion of male and female samples in training, from 100% to 0% males with a step of 10% between each sampling. Then, each data point of the distribution is interpolated to be between 1 and 100 years, adding a random noise of 10 years to simulate real cases of age regression.

### B.2  DIPDI on synthetic data: simulating model outputs

In this section, we verify the behaviour of DIPDI under controlled conditions using synthetic data by simulating model predictions. We simulate the predictions of two sets of models $\mathbb{A} = \{A_1, A_2\}$ and $\mathbb{B} = \{B_1, B_2\}$ when evaluated on samples from a synthetic dataset $\mathbb{D}$ and then systematically evaluate DIPDI in scenarios with different levels of disagreement between $\mathbb{A}$ and $\mathbb{B}$. The model discrepancy is here simulated by the addition of a stochastic value of varying size (disagreement level) to the output predictions (Figure 8a).

We consider the task of age estimation, so the outputs of models in $\mathbb{A}$ and $\mathbb{B}$ are assumed to represent *predicted ages*. We start with a fixed sample $\mathbb{Y}$ drawn from a uniform distribution of ages between 30 and 80, representing the *ground-truth ages*, $y_k \in \mathbb{Y}$. We simulate synthetic predictions for the models in $\mathbb{A}$ and $\mathbb{B}$ by perturbing $\mathbb{Y}$ with Gaussian noise sampled from distributions $n_{\mathbb{A}} \sim \mathcal{N}(0, \sigma_{\mathbb{A}})$ and $n_{\mathbb{B}} \sim \mathcal{N}(0, \sigma_{\mathbb{B}})$.

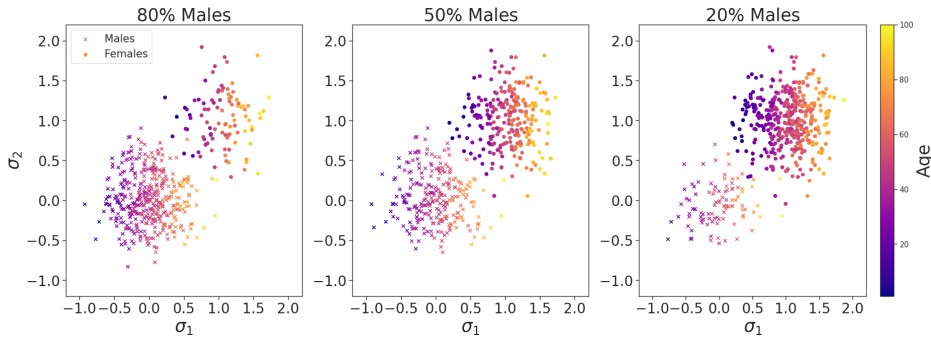

Figure 7: Examples of datasets generated for the synthetic experiment in Section 4.1.

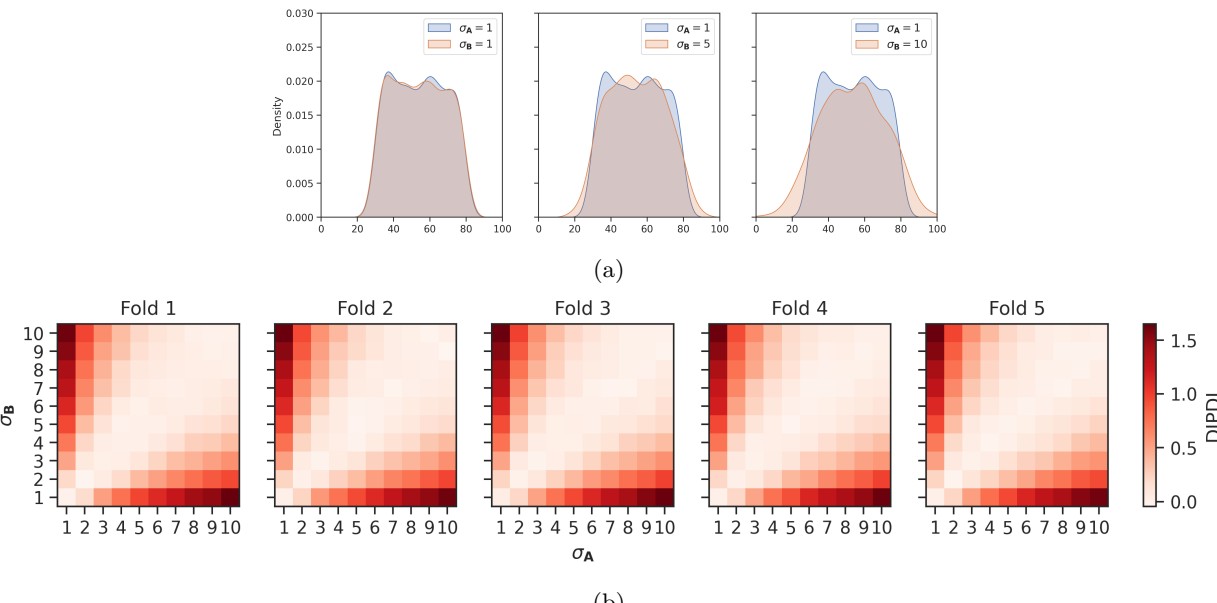

Figure 8: (a) Synthetic data construction. Examples of predicted ages simulated for models in $\mathbb{A}$ and $\mathbb{B}$, for increasing levels of prediction disparities (left to right). (b) DIPDI on synthetic data. Both $\sigma_{\mathbb{A}}$ and $\sigma_{\mathbb{B}}$ range from 1 to 10. Each fold represents a new run of the experiment with different random seeds. The DIPDI index is computed by averaging 3 simulations for each $\sigma_{\mathbb{A}}$.

Thus, for a fictitious data sample $k$ with ground-truth label $y_k \in \mathcal{Y}$, the synthetic model predictions are $A_i(\mathbf{x}_k) = y_k + n_{\mathbb{A}}$ and $B_i(\mathbf{x}_k) = y_k + n_{\mathbb{B}}$. Varying the standard deviations allows us to create scenarios where the predicted ages for the analysis groups are more or less similar, and then analyze the behaviour of DIPDI under different discrepancy ratios (see Figure 8a).

DIPDI values for different discrepancy scenarios are displayed in Figure 8b, considering $N = 1000$ and $\sigma_{\mathbb{A}}$ and $\sigma_{\mathbb{B}}$ values in the range [1-10]. Note that when the outputs of $\mathbb{A}$ and $\mathbb{B}$ are similarly perturbed (as shown on the diagonal of each image), then $\Phi$ is close to 0. However, when perturbations are sampled from a wider Gaussian in one set than the other (as shown outside the diagonal of each image), $\Phi$ tends to be higher than 0. This confirms the desired behaviour for our index: when intra-set predictions are more consistent than inter-set predictions, the index returns larger values.

## B.3 Subgroup analysis for age estimation

In this section, we present additional results in age estimation for ChestX-ray14 and UTKFace datasets. These results complement the insights discussed in Section 4.3 of the main manuscript.

Tables 2 and 3 present results for ChestX-ray14 and UTKFace, reporting the mean absolute error (MAE) values for male and female subgroups under different scenarios of gender imbalance in the training data.

| Training (Male-Female) | Testing on Male | Testing on Female |
|---|---|---|
| 100-0 | 4.633 (0.125) | 5.823 (0.301) |
| 75-25 | 4.679 (0.189) | 4.888 (0.109) |
| 50-50 | 4.795 (0.149) | 4.717 (0.149) |
| 25-75 | 4.802 (0.153) | 4.565 (0.105) |
| 0-100 | 5.265 (0.189) | 4.527 (0.109) |

Table 2: Mean absolute error (MAE) (mean ± std) for age estimation on ChestX-ray14 across subgroups (male, female) for models trained with different gender imbalance ratios.

| Training (Male-Female) | Testing on Male | Testing on Female |
|---|---|---|
| 100-0 | 6.170 (0.207) | 9.006 (0.442) |
| 75-25 | 6.301 (0.277) | 6.275 (0.368) |
| 50-50 | 6.530 (0.208) | 6.004 (0.319) |
| 25-75 | 6.785 (0.284) | 5.780 (0.326) |
| 0-100 | 7.468 (0.418) | 5.719 (0.427) |

Table 3: Mean absolute error (MAE) (mean ± std) for age estimation on UTKFace across subgroups (male, female) for models trained with different gender imbalance ratios.

Figure 9 shows the cumulative score (CS) values for ChestX-ray14 and UTKFace for male and female subgroups under different gender imbalance scenarios in the training data. The CS quantifies the proportion of test samples ($N$) for which the absolute error $e$ falls below a specified threshold of $n$ years. This calculation is defined as follows:

$$CS(n) = \frac{N_{e \leq n}}{N},$$

where $N_{e \leq n}$ represents the number of test images for which the absolute age error is less than or equal to the corresponding threshold value.

## B.4 Subgroup analysis for younger vs older classification

In this section, we present additional results regarding the classification of celebrities from the CelebA dataset in younger vs older. These results complement the insights discussed in Section 4.3 of the main manuscript.

Figure 10 shows normalized confusion matrices computed by subgroups (male, female) aggregating all folds for models trained with different gender imbalance ratios. Table 4 presents the Equality of Opportunity (EOD) metric (Hardt et al., 2016). These values, computed as the absolute difference over folds, reveal more unfair performance on younger vs older classification for models trained on highly imbalanced datasets, what is consistent with the behaviour of DIPDI shown in the results from the main manuscript (Figure 2).

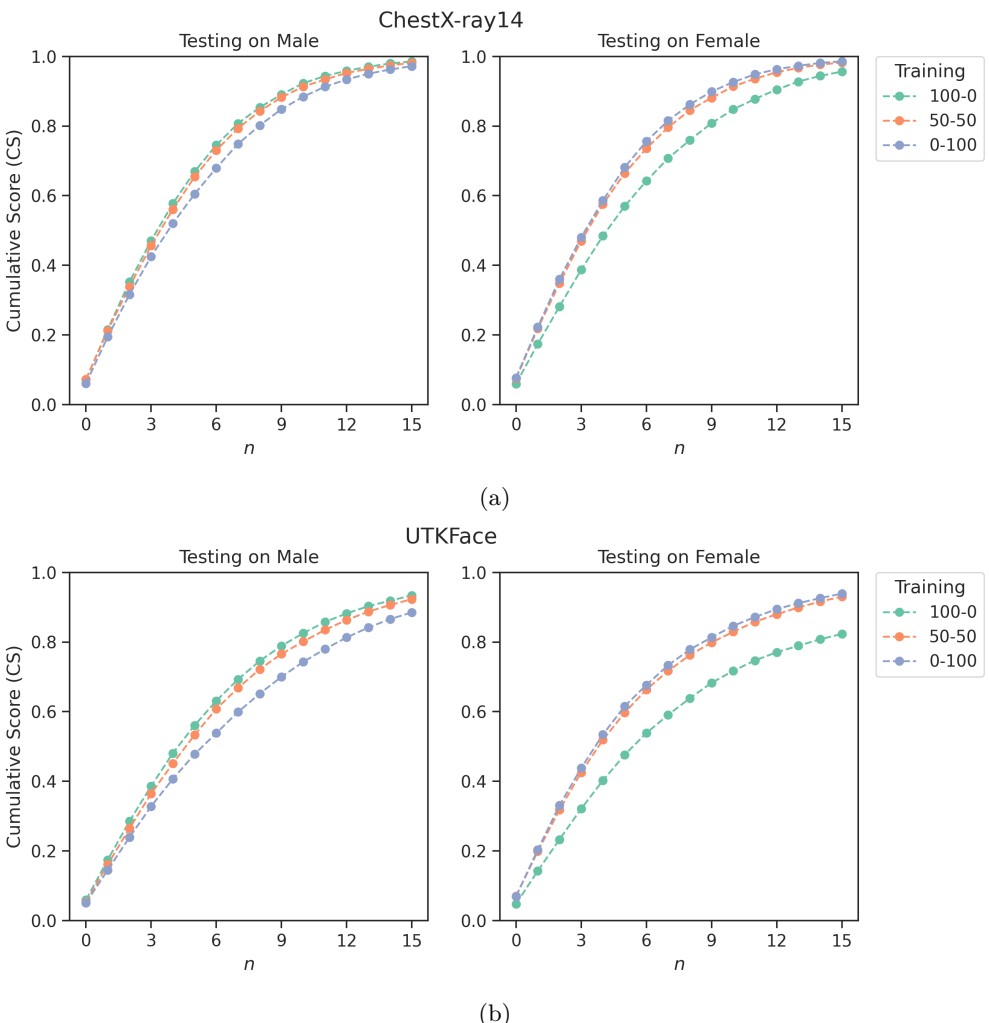

Figure 9: Cumulative scores (CS) for age estimation on ChestX-ray14 (a) and UTKFace (b) by subgroups (male, female) for models trained with different gender imbalance ratios. Age threshold $n$ spans from 0 to 15 years.

| Training (Male-Female) | EOD |
|:---:|:---:|
| 100-0 | 0.096 (0.043) |
| 75-25 | 0.030 (0.021) |
| 50-50 | 0.034 (0.025) |
| 25-75 | 0.052 (0.043) |
| 0-100 | 0.363 (0.071) |

Table 4: Equality of Opportunity (EOD) (mean ± std) for younger vs older classification on CelebA for models trained with different gender imbalance ratios.

## B.5   DIPDI for age estimation and younger vs older classification

Table 5 presents additional results for DIPDI on age estimation and younger vs older classification tasks as discussed in Section 4.4 of the main manuscript. We observe that DIPDI produces values larger than 0 when training data is highly imbalanced in gender attributes indicating a greater propensity to bias.

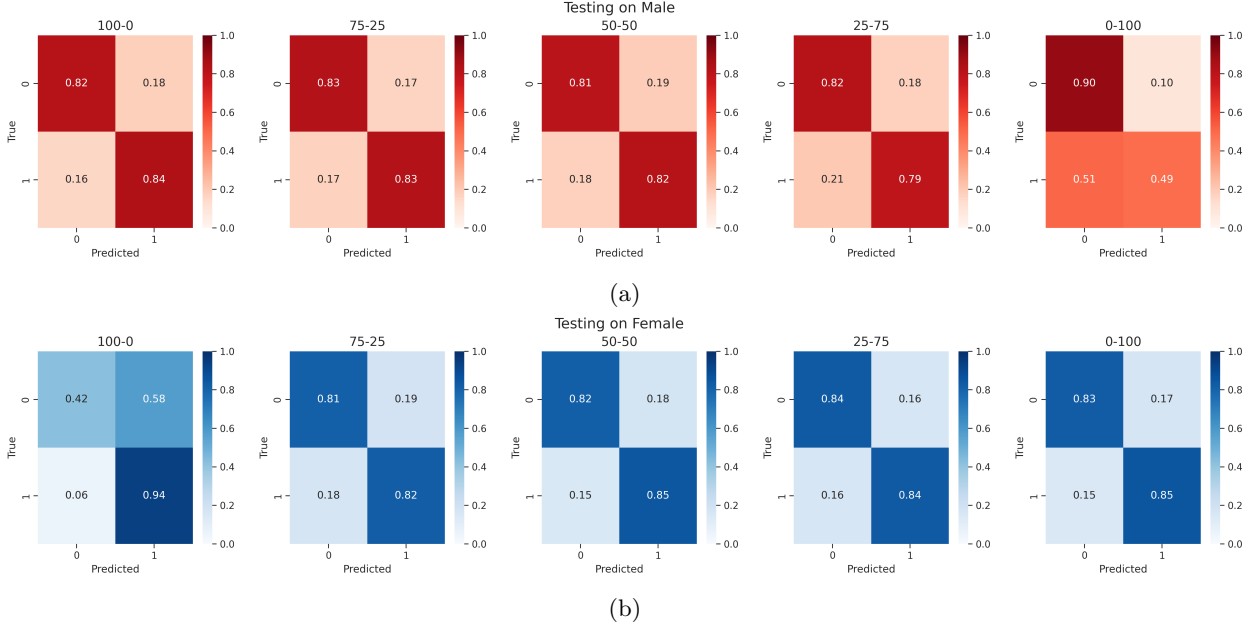

Figure 10: Normalized confusion matrices computed on male (a) and female (b) subgroups for models trained with different gender imbalance ratios.

| $\mathbb{A}$, $\mathbb{B}$ | ChestX-ray14 | UTKFace | IMDB-WIKI | CelebA |
|---|---|---|---|---|
| 100-0, 0-100 | 0.576 (0.103) | 0.918 (0.241) | 0.799 (0.271) | 1.447 (0.247) |
| 75-25, 25-75 | 0.027 (0.040) | 0.126 (0.063) | 0.273 (0.278) | 0.065 (0.167) |
| 50-50, 50-50 | -0.005 (0.065) | 0.010 (0.074) | -0.106 (0.224) | 0.014 (0.213) |
| 25-75, 75-25 | 0.012 (0.073) | 0.080 (0.050) | 0.558 (0.196) | 0.177 (0.163) |
| 0-100, 100-0 | 0.530 (0.114) | 0.883 (0.159) | 0.993 (0.346) | 1.501 (0.280) |

Table 5: DIPDI (mean ± std) for age estimation (ChestX-ray14, UTKFace and IMDB-WIKI) and younger vs older classification (CelebA). Groups $\mathbb{A}$ and $\mathbb{B}$ consist of two models trained with different gender imbalance ratios. Test data is gender balanced.

