# OpenReview forum: "Demographically-Informed Prediction Discrepancy Index: Early Warnings of Demographic Biases for Unlabeled Populations"
_TMLR — Accepted by TMLR_

### Review · Reviewer_EvUh · 2023-12-19

**Summary Of Contributions:**

EDIT: I have a couple of notes remaining in response but pending those being cleared up, I'm updated my review here to "Yes" on "Claims and Evidence", since the authors have updated their experimental section appropriately.

In this paper, the authors present a method for predicting when biases, in the form of disparities in accuracy, will arise at test time given unlabelled data. The method involves a metric, DIPDI, comparing the test-time difference in prediction between models that are trained solely on data from one sensitive group. The between group difference is compared to a within group difference, as determined by a sample of two models. Experimentally, the authors run their method on synthetic and real data and analyze the results, arguing that DIPDI successfully predicts disparities, even under particular kinds of distribution shift.

**Audience:**

Yes

**Claims And Evidence:**

No

**Requested Changes:**

My requests here are substantial enough that I don’t think I can qualify any as sufficient for securing my recommendation for acceptance. However, the proposed adjustments would include:

Most critical: Deepening and broadening of each of the 3 experimental sections as discussed above (synthetic: make the setup closer to the actual method, real data: more extensive exploration and analysis)
Also critical: clarification in the paper framing of what “bias” means throughout, narrowing scope to the specific metric of consideration
Moderate: explore the method in more conceptual depth, such as by drawing connections to distribution shift and by extending to larger ensembles than n=2
Moderate: extend the scope of experiments to outside only age prediction, or outside only images

I do think this idea is quite interesting and would enjoy seeing a more thorough treatment of it both conceptually and empirically.

**Strengths And Weaknesses:**

This is a compelling idea which I believe has the potential to draw on interesting connections across the fairness, robustness and distribution shift literature. I believe that the intuition behind this approach is sensible and could potentially provide useful test-time fairness signals, particularly in the case of shifted distributions. However, I’m not sure that the theoretical/conceptual work in this paper goes far enough to establish this idea in a fully-fleshed-out way, and I didn’t find that any of the experiments fully supported the claims made about the effectiveness of DIPDI.

Some feedback:
The main criticism I have around this paper is with the experiments. There are 3 experimental sections and I don’t necessarily find any of them particularly compelling as support for the claims made around DIPDI. I’ll go through these one at a time:
- Synthetic data - 4.1: I find these experiments unconvincing since they don’t really represent an application of the method; in particular, they skip a really key step which is the training of the models themselves. As currently set up it just seems like a demonstration of calculating the metric, which is not so helpful in my eyes. I think a good synthetic experiment which the full method (train a pool of models and compare outputs) is applied would be much more helpful.
- Real data IID - 4.2-4.4: I think these experiments are mostly fine but I found them a little underwhelming for 2 reasons. First of all, they are only conducted on 100-0 and 50-50 data imbalance; given how important data imbalance is to the paper I would have expected a slower gradation (show me 90-10, 80-20, etc.). This would be very helpful to show how sensitive the metric is. Secondly, I was disappointed to see that potential confounds were removed from the ChestX-ray14 set - handling these types of confounds is necessary in order for fairness metrics to perform well on real data. The authors should use the data as-is, or need to justify this choice in my opinion.
- Domain shift - 4.3: This is an interesting experiment and I’m glad the authors explored a shifted scenario. However, I’m confused by the main results shown in Figure 3. I’m not sure why these plots don’t look more similar between a and b. Specifically, it seems odd to me that the test=male subgroup doesn’t exhibit a similar diagonal pattern on the right hand side of 3a as the test=female subgroup does on the right hand side of 3b. This could use some more explanation as to why these behave differently and also as to the impact of this on the DIPDI metric. I also wouldn’t mind seeing more different experiments in distribution shift scenarios, possibly exploring different types - the claim that this metric is helpful under shift is intriguing but as of right now the experiments only address an extremely narrow version of that claim (attribute-specific label shift, with caveats).
- I think this paper misses out on some potentially strong and clarifying impact by not discussing the connection to distribution shift. In particular, I’d be curious to know in what cases we might expect this method to work (what are its limitations) and how that interacts with the nature of the distribution shift. The authors reference the Schrouff et al paper which I believe contains an example of this type of discussion. Tracing out the limitations here would be very helpful - as is, I’m left in the dark about where I might expect this method to work
- The claims around “bias” are unnecessarily broad - “bias” can entail many things and can imply many different metrics, however in this paper the authors are only concerned with disparities in accuracy. This is fine from a substance perspective - however I think that this should be mentioned prominently in the introduction and throughout, given the exclusive theoretical and empirical focus on that notion of bias.
- It’s unclear to me why the experiments are all age prediction - I don’t necessarily have a problem with it but I think given how general the proposed method is, it might behoove the authors to demonstrate its effectiveness on some other tasks. This would be even stronger in a non-image modality, but not required
- I think it would be interesting to see this metric defined in more generality - for instance, it seems like it could be made much stronger by extending to pools of > 2 models on each attribute. The authors don’t discuss this and as such I think the metric is more brittle than it needs to be





Smaller notes:
- Unclear exactly what a “new target population” is - is it a reweighting of existing sensitive groups? Reweighting along some other attribute? A totally new sensitive group? A different context?
- Should draw more connections to other literature like distribution shift and underspecification
- Citation format: missing brackets around many citations - look at \citep in LateX rather than \citet
- In 3.1, is D a dataset or an input space? Some notation confusion.
- In 3.1 why does the range of A have to be positive?
- I’m not sure I totally understand the discussion/analogy to uncertainty in the Discussion section

---

> ### Author Response · Authors · 2024-01-22
> **Answer to reviewer EvUh (part 1/5)**
>
> >This is a compelling idea which I believe has the potential to draw on interesting connections across the fairness, robustness and distribution shift literature. I believe that the intuition behind this approach is sensible and could potentially provide useful test-time fairness signals, particularly in the case of shifted distributions. However, I’m not sure that the theoretical/conceptual work in this paper goes far enough to establish this idea in a fully-fleshed-out way, and I didn’t find that any of the experiments fully supported the claims made about the effectiveness of DIPDI.
>
> >Some feedback: The main criticism I have around this paper is with the experiments. There are 3 experimental sections and I don’t necessarily find any of them particularly compelling as support for the claims made around DIPDI. I’ll go through these one at a time:
>
> We thank the reviewer for their comments. We have now significantly expanded the experimental section as requested and uploaded a revised PDF with the updated manuscript (changes are highlighted in blue). In what follows, we address every comment in detail.
>
> **Response to Weaknesses:**
>
> > 1.Synthetic data - 4.1: I find these experiments unconvincing since they don’t really represent an application of the method; in particular, they skip a really key step which is the training of the models themselves. As currently set up it just seems like a demonstration of calculating the metric, which is not so helpful in my eyes. I think a good synthetic experiment which the full method (train a pool of models and compare outputs) is applied would be much more helpful.
>
> → Thanks for this suggestion. Following your advice, we have now included a new controlled experiment (see Section 4.1 and Figure 1 from the updated manuscript), where instead of directly generating the output predictions, we actually generate synthetic datasets simulating different demographic groups for a simple regression task, train ML models on these samples, perform inference, and evaluate DIPDI on the actual predictions of these trained model. The results are consistent with our hypothesis that DIPDI helps to anticipate bias proneness as it correlates with the existence of bias in unseen data samples. For the sake of completeness, we have moved the previous simple example to the Appendix B.2.
>
> > 2. Real data IID - 4.2-4.4: I think these experiments are mostly fine but I found them a little underwhelming for 2 reasons. First of all, they are only conducted on 100-0 and 50-50 data imbalance; given how important data imbalance is to the paper I would have expected a slower gradation (show me 90-10, 80-20, etc.). This would be very helpful to show how sensitive the metric is.
>
> → We have now included intermediate graduations for 25-75 and 75-25 in all the experiments, which confirm the sensitivity of the metric also when intermediate imbalance ratios are used (see Figure 2).
>
> >3. Secondly, I was disappointed to see that potential confounds were removed from the ChestX-ray14 set - handling these types of confounds is necessary in order for fairness metrics to perform well on real data. The authors should use the data as-is, or need to justify this choice in my opinion.
>
> → We have now updated the experiments using the complete ChestX-ray14 dataset therefore using the dataset as is. Note that the same tendency is observed and conclusions still hold for this new configuration. The description of the experimental setup has been updated accordingly in Section 4.2, stating that our experiment now includes both healthy and pathological cases.
>
> (continue in the next comment ..)

---

> > ### Comment · Reviewer_EvUh · 2024-02-07
> > **Response**
> >
> > Thanks for these updates - taking a look through now. I'm not sure I see the update on the construction of the ChestX-ray14 dataset, where is that?

---

> ### Author Response · Authors · 2024-01-22
> **Answer to reviewer EvUh (part 2/5)**
>
> >4. Domain shift - 4.3: This is an interesting experiment and I’m glad the authors explored a shifted scenario. However, I’m confused by the main results shown in Figure 3. I’m not sure why these plots don’t look more similar between a and b. Specifically, it seems odd to me that the test=male subgroup doesn’t exhibit a similar diagonal pattern on the right hand side of 3a as the test=female subgroup does on the right hand side of 3b. This could use some more explanation as to why these behave differently and also as to the impact of this on the DIPDI metric.
>
> → Thanks for pointing this out. We agree that this is somehow counterintuitive and it could use some more explanation. To clarify, we would like to first remind that in this experiment, we are inducing distribution shifts (label shift in this case) in the test data by decreasing the subpopulation age of either female or male subjects, and we do it by increasing the proportion of subjects younger than 45 years old, from 50% to 90% for each gender separately. In principle, there could be three possibilities when compared with the original distribution: the problem is more, equally or less prone to be biased. To assess this, we estimate such bias proneness using the ground truth annotations (i.e. by measuring the MAE gap -ΔMAE- between models trained on the original distribution of male and female patients) and we observe if the estimated DIPDI is able to capture the same tendency. Thus, we first measure the gap in performance ΔMAE for models trained on male and female patients, and report it independently for testing on male (red) and female subjects (blue) in the right column of Figure 3 in the updated manuscript. When measuring the ΔMAE for both groups as we increase the proportion of younger males, such difference tends to slightly increase, and a corresponding slightly increase in DIPDI is observed (Figure 3.a, right panel), meaning that changing the age distribution for males slightly changes the bias proneness of the original models as measured using ground-truth labels. Importantly, the DIPDI index (computed without any annotations) reflects this fact, confirming our hypothesis.
>
> Moreover, when varying the proportion of females younger than 45 years old in the unseen population, we observe that the ΔMAE between models trained on male and female patients stays constant for males (red curve in Figure 3.b, right panel), but decreases for female subjects (blue curve), reaching a level of bias equivalent to the one observed for males (at 90%, where the blue and red curves intersect). In other words, decreasing the age of the female population changes the bias proneness of that group as measured by analyzing the ΔMAEs, making it more fair as it reaches levels for bias proneness equivalent in both populations. As expected, the DIPDI index follows exactly the same tendency (is reduced as bias proneness is reduced), confirming our hypothesis.
>
> Regarding the different behavior observed when introducing label shifts for test=male and test=female subgroups, we hypothesize that it may be rooted in the different ways in which the features of each subgroup interact with age. In fact, for this dataset it is well known that the baseline performance is different for both subgroups. This would explain the greater susceptibility of a subgroup to changes in age distribution. Interestingly, as the DIPDI is calculated from the dispersion of scores throughout the models, and the underlying hypothesis is that these dispersions increase when the performance of a group decreases, if performance of test=male is not significantly affected by the shift then the dispersion will not increase, and therefore the DIPDI will remain mostly unchanged.
>
> This is now better discussed in the second paragraph of Section 4.5.1.
>
> (continue in next comment)

---

> ### Author Response · Authors · 2024-01-22
> **Answer to reviewer EvUh (part 3/5)**
>
> > 5. I also wouldn’t mind seeing more different experiments in distribution shift scenarios, possibly exploring different types - the claim that this metric is helpful under shift is intriguing but as of right now the experiments only address an extremely narrow version of that claim (attribute-specific label shift, with caveats).
>
> → We thank the reviewer for the suggestion. We have now included additional experiments for a different type of domain shift in a different dataset and task. Instead of focusing solely on regression, we now also evaluate a classification model which is trained to distinguish between blond and non-blond celebrities in the CelebA dataset. As shown in the new experiment, this task is prone to be biased with respect to gender, and DIPDI is clearly capturing this. In the additional domain shift scenario that we include now (Figure 4), we produce a different type of distribution shift, by transforming all the color test images into grayscale images. When evaluating bias of the original models in the shifted distribution using the ground-truth annotations, we note that the problem becomes less prone to be biased overall as the gap in accuracy (ΔAccuracy) between models trained on male and female subjects becomes more similar  in grayscale than RGB images. Importantly, DIPDI consistently captures this behavior as it decreases when computed using the grayscale images in comparison to the RGB case. A complete discussion of this experiment is now included in Section 4.5.2.
>
> >6. I think this paper misses out on some potentially strong and clarifying impact by not discussing the connection to distribution shift. In particular, I’d be curious to know in what cases we might expect this method to work (what are its limitations) and how that interacts with the nature of the distribution shift. The authors reference the Schrouff et al paper which I believe contains an example of this type of discussion. Tracing out the limitations here would be very helpful - as is, I’m left in the dark about where I might expect this method to work
>
> → As previously mentioned, we have now expanded the experimental validation related to distribution shifts by including a new type of domain shift (RGB to grayscale) in a new problem, namely classification of blond vs non-blond celebrities in the CelebA dataset (see Section 4.5.2). Moreover, following the suggestion of the reviewer about the discussion presented in Schrouff et al 2022, we now discuss the particular type of distribution shifts that we are dealing with (namely label and covariate distribution shifts), mentioning it as a limitation and leaving the validation of DIPDI to measure other types of domain shift as future work.
>
> > 7. The claims around “bias” are unnecessarily broad - “bias” can entail many things and can imply many different metrics, however in this paper the authors are only concerned with disparities in accuracy. This is fine from a substance perspective - however I think that this should be mentioned prominently in the introduction and throughout, given the exclusive theoretical and empirical focus on that notion of bias.
>
> → Following the reviewer's comment, we narrowed down the definition of bias used in this study. Please see the 2nd paragraph of the Introduction section. We have also made minor modifications in the rest of the text to ensure this is clear throughout the manuscript. Particularly note the modification done to the 'Contributions' paragraph in the Introduction section, where we stress this fact.
>
> >8. It’s unclear to me why the experiments are all age prediction - I don’t necessarily have a problem with it but I think given how general the proposed method is, it might behoove the authors to demonstrate its effectiveness on some other tasks. This would be even stronger in a non-image modality, but not required
>
> → We have now expanded the experimental validation including classification scenarios. Instead of focusing solely on regression, we now also evaluate classification models using the CelebA dataset. We evaluate two different tasks: first, classifying celebrities as younger vs older, posing the problem as a binary classification problem (see Figure 2.g-h). To this end, we also extend the definition of DIPDI to classification problems, using the Jensen-Shannon divergence as the discrepancy function. Moreover, we also evaluate a distribution shift scenario for a classification problem. In this case, the problem is to distinguish between blond and non-blond celebrities in the CelebA dataset, and we introduce a covariate shift by changing from color to grayscale images (see Section 4.5.2).

---

> > ### Author Response · Authors · 2024-01-22
> > **Answer to reviewer EvUh (part 4/5)**
> >
> > >9. I think it would be interesting to see this metric defined in more generality - for instance, it seems like it could be made much stronger by extending to pools of > 2 models on each attribute. The authors don’t discuss this and as such I think the metric is more brittle than it needs to be
> >
> > → We have now included a clear explanation about how DIPDI can be extended for pools of more than 2 models. To keep the main text simple, we included this formulation as a new section in the Appendix A.1.
> >
> > **Response to Smaller notes:**
> >
> > >10. Unclear exactly what a “new target population” is - is it a reweighting of existing sensitive groups? Reweighting along some other attribute? A totally new sensitive group? A different context?
> >
> > → When we use the term 'new target population', we refer to a new set of individuals that was not seen during training, that may have suffered a distribution shift or not. In our experiments we include two types of distribution shift for the new target population: label shift (by changing the age distribution of one of the subgroups) and image intensity shift (we go from RGB to grayscale images), showing that DIPDI can capture the changes in bias proneness that these distribution shifts introduce. This term, which was used in the abstract, was replaced by "unseen population" which is more descriptive.
> >
> > >11. Should draw more connections to other literature like distribution shift and underspecification
> >
> > → We have now discussed and adopted the terminology from Schrouff et al 2022 to refer to the type of distribution shifts we are analyzing (namely covariate and label shifts) and included a new experiment showing how DIPDI can capture changes in bias proneness due to covariate shifts (in our case, going from color to grayscale images), that was not part of the initial submission.
> >
> > >12. Citation format: missing brackets around many citations - look at \citep in LateX rather than \citet
> >
> > → We have changed it as requested.
> >
> > >13. In 3.1, is D a dataset or an input space? Some notation confusion.
> >
> > → D is a dataset. We have now changed the notation to avoid confusion.
> >
> > >14. In 3.1 why does the range of A have to be positive?
> >
> > → Thanks for pointing this out. Initially we restricted it to positive real numbers because we were focusing on the age regression task. But now we have formulated it in a more general way and removed the positive constraint.
> >
> > >15. I’m not sure I totally understand the discussion/analogy to uncertainty in the Discussion section
> >
> > → We believe this is an interesting analogy as it may spark new ideas by exploring the relation between model uncertainty and bias proneness. In particular, uncertainty in ensemble models is usually estimated as the variance in the predictions of components of the ensemble: if all models in an ensemble agree on a prediction, the uncertainty for this sample is likely low. Conversely, if there is a high variance in the predictions across the models, this indicates higher uncertainty. When we evaluate the discrepancy in the predictions of models trained for a particular demographic group, we could interpret them as models in an ensemble, and consequently, the discrepancy in their predictions for a given subject could be seen as the uncertainty of such an ensemble. In that sense, our index quantifies the uncertainty estimated using 'ensembles' of models trained with data from different demographic groups (numerator), normalized by models trained from the same demographic group (denominator). If both are similar, then we get a DIPDI value close to 0, indicating that the problem shows no early signs of potential bias with respect to the analyzed demographic values. However, higher discrepancies for models from different demographic groups than for models from the same demographic group will lead to DIPDI values significantly larger than 0, indicating bias proneness for the task under analysis. We have modified the 'Discussion' section to better reflect this clarification.

---

> > > ### Author Response · Authors · 2024-01-22
> > > **Answer to reviewer EvUh (part 5/5)**
> > >
> > > **Response to Requested Changes:**
> > >
> > > > My requests here are substantial enough that I don’t think I can qualify any as sufficient for securing my recommendation for acceptance. However, the proposed adjustments would include:
> > >
> > > > Most critical: Deepening and broadening of each of the 3 experimental sections as discussed above (synthetic: make the setup closer to the actual method, real data: more extensive exploration and analysis)
> > >
> > > → We have considerably extended the experimental section following the reviewer's advice, using both synthetic and real data. We refer the reviewer to points 1, 2, 3, 5, and 8, where we discuss the additional experiments added to the revised manuscript. In brief, they include:
> > > - A more realistic synthetic experiment with a setup closer to the actual method (Section 4.1).
> > > - Additional experiments for intermediate imbalance ratios (25-75% and 75-25%) when analyzing DIPDI (Figure 2).
> > > - Redoing the experiment using the real distribution of ChestX-ray14 which includes healthy and pathological cases (Figure 2.a and 2.b).
> > > - The extension of the DIPDI index for classification problems and its validation using a new dataset and task (Section 4.3).
> > > - A new experiment covering an additional type of distribution shift (covariate shift, Section 4.5.2).
> > >
> > > > Also critical: clarification in the paper framing of what “bias” means throughout, narrowing scope to the specific metric of consideration
> > >
> > > → We narrowed down the definition of bias used in this study. Please see the answer to point 7 where we detail this modification.
> > >
> > > >Moderate: explore the method in more conceptual depth, such as by drawing connections to distribution shift and by extending to larger ensembles than n=2
> > >
> > > → We now include an extension of the DIPDI formulation considering pools of more than two models in Appendix A.1. Moreover, we provide a theoretical derivation of analytic expressions for the relationship between performance gap and output discrepancies captured by DIPDI (Appendix A.2). Finally, we have expanded the discussion about distribution shifts and DIPDI by including experiments about covariate and label shifts and linking it to Schrouff et al, 2022 as explained in point 11.
> > >
> > > >Moderate: extend the scope of experiments to outside only age prediction, or outside only images
> > >
> > > → As previously mentioned, we extended the experimental validation to include classification problems (namely hair color classification and classifying younger vs. older individuals), and we also included a new distribution shift experiment.
> > >
> > > > I do think this idea is quite interesting and would enjoy seeing a more thorough treatment of it both conceptually and empirically.
> > >
> > > →  We thank the reviewer for their comments and positive feedback. We believe the extended experimental validation (both in real and synthetic datasets), together with the new theoretical derivations and additional discussions, have substantially improved our manuscript.

---

> > > > ### Comment · Reviewer_EvUh · 2024-02-07
> > > > **Final Response**
> > > >
> > > > Thanks for these updates. It looks like you've addressed my main issues with the paper and assuming that my main questions below are addressed (chest xray clarification, citation format), happy to recommend this for acceptance.

---

> > > > > ### Author Response · Authors · 2024-02-09
> > > > > **Answer to reviewer EvUh**
> > > > >
> > > > > → Thanks to the reviewer for their positive feedback. We have now updated the revised manuscript addressing citation and notation issues. Below, we present a response to their last comments:
> > > > >
> > > > > > 1. Thanks for these updates - taking a look through now. I'm not sure I see the update on the construction of the ChestX-ray14 dataset, where is that?
> > > > >
> > > > > → The following paragraph in Section 4.2 on page 7 refers to the construction of the dataset. In particular, the sentence in bold clarifies the fact that we have included both healthy and pathological cases as requested: "The ChestX-ray14 dataset contains 112,120 high-resolution frontal-view radiographs of 30,805 unique patients with age and gender labels. Each image is annotated with up to 14 different chest disease labels extracted from radiology reports (not used in our study), age and gender.  We use the ChestX-ray14 dataset to perform subgroup analysis in terms of gender and to evaluate DIPDI in models trained to perform age estimation from radiological images. We use here for gender the binary labels reported in the dataset,  i.e. male and female. **To avoid having two images of the same patient in train and test, we randomly selected one image per patient, resulting in a total of 28,350 images which include healthy and pathological cases.**”
> > > > >
> > > > > > 2. On the general ensemble formulation of DIPDI - it's not clear to me why in the m=4 case this is the form this would take. Why wouldn't we look at instead the product of all possible intra-set pairs in the denominator? Is this just a computational restriction or is there another reason?
> > > > >
> > > > > → This is due to two reasons: on the one side, as mentioned by the reviewer, due to computational efficiency. On the other hand, our approach guarantees a fair comparison between intra and inter-set pairs, with the same statistical power, as the formulation results in the numerator and the denominator always containing the same number of pairs. This guarantees that all models from A and B participate equally in the computations and therefore allows us to express DIPDI for other values of m as an average of DIPDIs with m=2, which would not be possible if we considered all possible pairs. Moreover, the formulation needs to guarantee that when DIPDI is evaluated in a setting where A and B are actually the same distribution of models, they are still different models from the distribution, and no comparison of a model with itself is computed.
> > > > >
> > > > > > 3. thank you for the citation formatting update! I see the brackets in many now. However, I want to clarify I wasn't asking for all the citet to be replaced with citep. Rather, they are different commands that can be used in different situations to improve readability. The no-brackets version (citet) is used when the author's name is intended to be part of the sentence (e.g. In Petersen et al., (2022) for example, the authors found that in the case of Alzheimer’s disease prediction from brain magnetic resonance images (MRI)). The brackets version (citep) is intended to be used when it's attached to the sentence as a pseudo-footnote (e.g. Model performance across demographic groups is commonly evaluated employing one or more metrics (Corbett-Davies & Goel, 2018) with the implicit assumption...). Getting these appropriately matched up really improves readability and I think should be corrected before final submission
> > > > >
> > > > > → This was an unintentional error and we thank the reviewer for pointing it out. We have now corrected the citation format in the new manuscript.
> > > > >
> > > > > > 4. on notation: A: D -> Y still implies D is the domain of X rather than the dataset, I'd change this to like D_x or \mathds{X} or something.
> > > > >
> > > > > → We have changed the notation to clearly distinguish the input domain (now referred to as X) from the dataset (D).

---

> > > ### Comment · Reviewer_EvUh · 2024-02-07
> > > **Response 2**
> > >
> > > - On the general ensemble formulation of DIPDI - it's not clear to me why in the m=4 case this is the form this would take. Why wouldn't we look at instead the product of all possible intra-set pairs in the denominator? Is this just a computational restriction or is there another reason?
> > >
> > > - thank you for the citation formatting update! I see the brackets in many now. However, I want to clarify I wasn't asking for all the citet to be replaced with citep. Rather, they are different commands that can be used in different situations to improve readability. The no-brackets version (citet) is used when the author's name is intended to be part of the sentence (e.g. In Petersen et al., (2022) for example, the authors found that in the case of Alzheimer’s disease prediction
> > > from brain magnetic resonance images (MRI)). The brackets version (citep) is intended to be used when it's attached to the sentence as a pseudo-footnote (e.g.  Model performance across demographic groups is commonly evaluated employing
> > > one or more metrics (Corbett-Davies & Goel, 2018) with the implicit assumption...). Getting these appropriately matched up really improves readability and I think should be corrected before final submission
> > >
> > > - on notation: A: D -> Y still implies D is the domain of X rather than the dataset, I'd change this to like D_x or \mathds{X} or something.

---

### Review · Reviewer_iDM1 · 2023-12-31

**Summary Of Contributions:**

Data imbalance on sub-groups in a dataset may or may not lead to unequal performance.
This paper proposes a method to detect biases related to a demographic attribute in unlabeled datasets. The proposed approach involves analyzing output consistency among models trained on different demographic groups. Using a novel index, DIPDI, the authors assess whether models exhibit inconsistent predictions for the same data, indicating potential bias against a specific demographic attribute. Validation on synthetic and real-world datasets, specifically age estimation from face photos and X-ray images, demonstrates DIPDI's ability to anticipate bias without ground truth labels. The metric is effective in assessing bias in dynamic contexts with demographic shifts and provides early warnings in real-world scenarios without requiring expert annotations.

**Audience:**

Yes

**Broader Impact Concerns:**

Part of the wordings in the final section “Discussion” could be added as a paragraph for the broader impacts.

**Claims And Evidence:**

No

**Requested Changes:**

1. It is recommended that the authors provide visualization or simple numerical examples in Section 3 to illustrate the intuition of the proposed metric DIPDI, despite that the definition is simple. E.g., concrete examples for functions in set \mathbb{A} and \mathbb{B}. Why there are only two models? Is DIPDI able to be extended to many models in the sets?
2. The definition of DIPDI is formed for regression problems, it is recommended that the authors provide an according definition of DIPDI for classification tasks, and compare this metric with existing fairness metrics in order to add value to this work.
3. Why choosing 50-50, 0-100 for the models in Table 1? Why not changing from 0 to 100 with a difference of 10, i.e., 10 different settings?
4. Can the authors add a simple theoretical analysis between the relationships of distribution shift and DIPDI using some simple examples?
5. The authors claim that the DIPDI can find potential bias. Can the authors validate it with a dataset that has ground-truth annotation to check the usefulness of this new metric?
6. How do we related the definition of DIPDI with counter-factual desperate impacts?

**Strengths And Weaknesses:**

1. The survey of past literature is comprehensive.
2. The authors provide abundant experiments with not only synthetic but also three real-world datasets.
For weaknesses, please refer to “Requested Changes.”

---

> ### Author Response · Authors · 2024-01-22
> **Answer to reviewer iDM1 (part 1/2)**
>
> Thanks to the reviewer for their comments. We have uploaded a revised manuscript addressing their concerns. Below, we present a detailed response to the requested changes:
>
> > 1. It is recommended that the authors provide visualization or simple numerical examples in Section 3 to illustrate the intuition of the proposed metric DIPDI, despite that the definition is simple. E.g., concrete examples for functions in set \mathbb{A} and \mathbb{B}. Why there are only two models? Is DIPDI able to be extended to many models in the sets?
>
> → We have now improved the synthetic experiment, including a new simple numerical experiment (see Section 4.1 and Figure 1 in the updated manuscript), where instead of directly generating the output predictions, we actually generate synthetic datasets simulating different demographic groups for a simple regression task, train ML models on these samples, perform inference, and evaluate DIPDI on the actual predictions of these trained model. We include a clear visualization of the resulting dataset in the new Appendix B.1. For the sake of completeness, we have moved the previous synthetic example to the Appendix B.2.
>
> Regarding the extension of DIPDI to more than two models, we now include a clear explanation about how to perform such extension. To keep the main text simple, we included this formulation as a new section in the Appendix A.1.
>
> > 2. The definition of DIPDI is formed for regression problems, it is recommended that the authors provide an according definition of DIPDI for classification tasks, and compare this metric with existing fairness metrics in order to add value to this work.
>
> → We have now expanded the experimental validation including classification scenarios. Instead of focusing solely on regression, we now also evaluate classification models using the CelebA dataset. We evaluate two different tasks: first, classifying celebrities as younger vs. older, posing the problem as a binary classification problem (see Figure 2.g-h). To this end, we also extend the definition of DIPDI to classification problems, using the Jensen-Shannon divergence as discrepancy function. Moreover, we also evaluate a distribution shift scenario for a classification problem. In this case, the problem is to distinguish between blond and non-blond celebrities in the CelebA dataset, and we introduce a covariate shift by changing from color to grayscale images (see Section 4.5.2).
>
> Regarding other metrics, in the new Appendix B.4 we include results when computing the fairness metric "Equality of Opportunity", usually employed in classification scenarios.
>
> Finally, it is worth noting that we modified Section 3.1 making the DIPDI formulation more generic, so that it can be easily instantiated for regression or classification problems.
>
> > 3. Why choosing 50-50, 0-100 for the models in Table 1? Why not changing from 0 to 100 with a difference of 10, i.e., 10 different settings?
>
> → We have now included intermediate graduations for 25-75 and 75-25 in all the experiments, which confirm the sensitivity of the metric also when intermediate imbalance ratios are used (see the new Figure 2).
>
> > 4. Can the authors add a simple theoretical analysis between the relationships of distribution shift and DIPDI using some simple examples?
>
> → We include a new theoretical analysis in Appendix A.2, computing DIPDI for simple examples and analyzing the relationship between DIPDI and performance gap.
>
> > 5. The authors claim that the DIPDI can find potential bias. Can the authors validate it with a dataset that has ground-truth annotation to check the usefulness of this new metric?
>
> → In the updated manuscript, we provide validation for 4 real datasets with ground-truth annotations, considering the ChestX-ray14 dataset, IMDB-Wiki and UTKFace (which includes ground-truth annotations for age as well as metadata for gender) and the CelebA dataset (which includes ground-truth annotations for age and hair color, as well as metadata for gender).
>
> (continue in next comment ...)

---

> > ### Author Response · Authors · 2024-01-22
> > **Answer to reviewer iDM1 (part 2/2)**
> >
> > > 6. How do we related the definition of DIPDI with counter-factual desperate impacts?
> >
> > → There is indeed an interesting relationship between DIPDI and counter-factual disparate impacts. In that sense, the concept of counter-factual disparate impact refers to the hypothetical impact of a predictive model or decision-making process on protected groups under counterfactual scenarios. It essentially assesses how outcomes would differ if certain protected attributes (like gender or race) were different, holding all other variables constant. By comparing the consistency in the predictions of models trained separately on different demographic groups, DIPDI implicitly engages in a form of counterfactual analysis. It assesses how model predictions would differ if the demographic attribute for the training sets were different, given the same test input data.  In the context of counterfactual disparate impact, the discrepancy in predictions made by the demographically informed models (e.g. male vs. female) can highlight areas where outcomes might significantly differ based on the protected attribute. DIPDI builds on top of the idea that such discrepancy is a proxy for bias proneness, which might be observed in real-world scenarios where predictions are made based on these models.

---

### Review · Reviewer_AMxc · 2024-01-08

**Summary Of Contributions:**

This research introduces the Demographically-Informed Prediction Discrepancy Index (DIPDI) as a method to anticipate potential demographic biases in machine learning models when deploying on new and unlabelled populations.

**Audience:**

Yes

**Broader Impact Concerns:**

A Broader Impact Statement on the ethical implications is required in this work.

**Claims And Evidence:**

No

**Requested Changes:**

Please refer to the weaknesses above.

**Strengths And Weaknesses:**

Strengths:
- The paper addresses the issue of anticipating potential demographic biases at deployment in the absence of ground truth annotations.

- The proposed Demographically-Informed Prediction Discrepancy Index (DIPDI) provides a measure for the proneness towards biased solutions and can serve as an early warning about potential demographic biases.

- The paper demonstrates the applicability of DIPDI in understanding how fairness transfers across distributions, particularly in scenarios involving population shifts.

Weaknesses:
- This paper lacks the theoretical results on the relationship between the index Demographically-Informed Prediction Discrepancy Index (DIPDI) and demographic biases.

- Could DIPDI also be used as the potential bias measurement for other existing statistical fairness notions? Besides, is there a comparation between DIPDI with existing fairness assessment methods?

- DIPDI is limited as a global population average. This paper does not explore its application to population subsets defined by multiple demographic traits or on a subject-by-subject basis.

---

> ### Author Response · Authors · 2024-01-22
> **Answer to reviewer AMxc**
>
> We thank the reviewer for their comments. We uploaded a revised version of the manuscript where we address their concerns, and here we include a point-by-point response to the weaknesses and requested changes:
>
> > This paper lacks the theoretical results on the relationship between the index Demographically-Informed Prediction Discrepancy Index (DIPDI) and demographic biases.
>
> →Thanks for the suggestion. We now include a new theoretical analysis section in Appendix A.2, analyzing the relationship between DIPDI and demographic biases as measured by performance gap.
>
> > Could DIPDI also be used as the potential bias measurement for other existing statistical fairness notions? Besides, is there a comparison between DIPDI with existing fairness assessment methods?
>
> → DIPDI bears a relation to existing standard definitions of fairness. With respect to classical group fairness, DIPDI could complement traditional group fairness metrics like equality of opportunity or gaps in performance between demographic groups, by providing additional insights into how consistent such predictions are across different demographic groups, particularly when ground-truth labels are not available. In fact, our experimental analysis shows that DIPDI can capture biases that are also reflected in standard metrics like the gap in MAE for regression models (Figure 2), as well as the gap in accuracy (Figure 2) and Equality of Opportunity (Table 4 in Appendix B.4) for classification problems.
>
> As now mentioned in the last paragraph of the Discussion section, while we have expressed DIPDI here as a global population average, the same reasoning could in principle be applied to population subsets defined by the intersection of multiple demographic traits (i.e. intersectional fairness), or even on a subject-by-subject basis, closer to the definition of individual fairness. Regarding counterfactual fairness approaches, DIPDI shares some resemblance as it involves a form of hypothetical scenario analysis. However, DIPDI's approach is more empirical, focusing on the discrepancies in predictions of actual models trained on different populations, while counterfactual fairness measures often involve more complex causal modeling and assumptions. Finally, in relation to approaches based on fairness through unawareness, which simply exclude protected attributes from the model, DIPDI actively measures the impact of these attributes by training separate models on different demographic groups. Overall, we believe that DIPDI offers a fresh perspective in the fairness literature, focusing on the unsupervised setting, which is not commonly discussed in this field, and may spark new discussions towards developing novel unsupervised bias discovery methods to anticipate bias issues in the absence of ground-truth.
>
> We have now updated the Discussion section to reflect these ideas.
>
> > DIPDI is limited as a global population average. This paper does not explore its application to population subsets defined by multiple demographic traits or on a subject-by-subject basis.
>
> → Here we focus on group fairness with exclusive consideration of a single sensitive attribute. However, we believe that the applicability of DIPDI extends beyond this specific context, and could potentially be explored for groups characterized by multiple attributes (intersectional fairness) or even at an individual level (individual fairness). This venue represents a direction for future research, and we have briefly mentioned it in the last paragraph of the Discussion section.

---

### Author Response · Authors · 2024-01-22
**Answer to reviewers and revised manuscript uploaded**

We thank the reviewers for their valuable comments. We are submitting a revised version of the paper addressing their comments, where changes are highlighted in blue. Importantly, we provide a point-by-point response as an answer to every reviewer's comment.

The most significant changes include:

- A more realistic synthetic experiment with a setup closer to the actual method
- Additional experiments for intermediate imbalance ratios (25-75% and 75-25%) when analyzing DIPDI
- The extension of the DIPDI index for classification problems and its validation using a new dataset and task
- A new experiment covering an additional type of distribution shift (covariate shift)
- A theoretical derivation discussing the relationship between DIPDI and performance gap
- A generalized definition of DIPDI for an arbitrary number of models

Overall, we have addressed both major and minor comments raised by the 3 reviewers, improving the quality of the final manuscript.

---

### Decision · Action_Editor_gxBQ · 2024-02-17

**Recommendation:** Accept as is

**Comment:**

This study examines how machine learning systems can exhibit biases against certain demographic groups due to data imbalance and under-representation in training datasets. The authors propose an approach to identify potential biases using output discrepancy among models trained on different demographic groups. The proposed method DIPDI measures the consistency of model predictions without resorting to the labels in the new target population. Through numerical experiments with synthetic and real-world datasets, they demonstrate how DIPDI can help anticipate demographic biases in machine learning models deployed on new and unlabeled populations.

The reviewers agree that the paper solves an important problem in fair machine learning, the proposed method is well motivated and reasonable. The reviewers also suggested adding more theoretical justifications and experiments to illustrate the effectiveness of the method.  After revision, two reviewers are satisfied with the new version. Reviewer AMxc still has some concerns regarding the theoretical analysis and empirical evidence. However, both myself and the other two reviewers believe that the revised version now includes more comprehensive theoretical and empirical analysis, meeting the criteria for publication. Overall, I think the paper proposes a new method and verifies its effectiveness and the presentation is clear. I thus recommend for acceptance.

**Audience:**

Yes

**Claims And Evidence:**

Yes